# Targeting the mevalonate or Wnt pathways to overcome CAR T-cell resistance in *TP53*-mutant AML cells

Jan Mueller[1,5], Roman R Schimmer [1,5], Christian Koch [1], Florin Schneiter[2], Jonas Fullin[1], Veronika Lysenko[1], Christian Pellegrino [1], Nancy Klemm[1], Norman Russkamp[1], Renier Myburgh[1], Laura Volta [1], Alexandre PA Theocharides[1], Kari J Kurppa [3], Benjamin L Ebert[4], Timm Schroeder[2], Markus G Manz [1,6] & Steffen Boettcher [1,6]✉

## Abstract

*TP53*-mutant acute myeloid leukemia (AML) and myelodysplastic neoplasms (MDS) are characterized by chemotherapy resistance and represent an unmet clinical need. Chimeric antigen receptor (CAR) T-cells might be a promising therapeutic option for *TP53*-mutant AML/MDS. However, the impact of *TP53* deficiency in AML cells on the efficacy of CAR T-cells is unknown. We here show that CAR T-cells engaging *TP53*-deficient leukemia cells exhibit a prolonged interaction time, upregulate exhaustion markers, and are inefficient to control AML cell outgrowth in vitro and in vivo compared to *TP53* wild-type cells. Transcriptional profiling revealed that the mevalonate pathway is upregulated in *TP53*-deficient AML cells under CAR T-cell attack, while CAR T-cells engaging *TP53*-deficient AML cells downregulate the Wnt pathway. In vitro rational targeting of either of these pathways rescues AML cell sensitivity to CAR T-cell-mediated killing. We thus demonstrate that *TP53* deficiency confers resistance to CAR T-cell therapy and identify the mevalonate pathway as a therapeutic vulnerability of *TP53*-deficient AML cells engaged by CAR T-cells, and the Wnt pathway as a promising CAR T-cell therapy-enhancing approach for *TP53*-deficient AML/MDS.

**Keywords** *TP53* Mutations; CAR T-Cell Therapy; AML; Mevalonate Pathway
**Subject Categories** Cancer; Haematology; Immunology

## Introduction

Acute myeloid leukemia (AML) and myelodysplastic syndromes (MDS) harboring *TP53* mutations or deletions are distinct clinicogenomic entities (Arber et al, 2022; Döhner et al, 2022; Khoury et al, 2022). They are associated with therapy resistance to conventional induction chemotherapy as well as hypomethylating agents (HMAs) with or without the BCL-2 inhibitor venetoclax (Ven) (Pollyea et al, 2022; Rucker et al, 2012). Even after allogeneic hematopoietic stem cell transplantation relapse rates are high (Lindsley et al, 2017; Middeke et al, 2016). Consequently, patients with *TP53*-mutant AML/MDS have extremely poor outcomes, and, thus, represent a major unmet clinical need (Bejar et al, 2011; Daver et al, 2022; Papaemmanuil et al, 2016; Rucker et al, 2012).

p53—encoded by its gene *TP53* on chromosome 17—is a transcription factor acting as a tumor suppressor by mediating the response to a variety of cellular stress stimuli such as DNA damage and other pro-apoptotic signals (Levine and Oren, 2009). Current treatment options for AML/MDS, including conventional induction chemotherapy and HMAs +/− Ven, largely exert their efficacy via the induction of p53-mediated apoptosis (Schimmer et al, 2022; Thijssen et al, 2021). Therefore, novel therapeutic strategies—in particular immunotherapeutic approaches that possibly circumvent the need for p53-mediated apoptosis—appear particularly promising for this hard-to-treat patient population (Sallman et al, 2020). Chimeric antigen receptor (CAR) T-cell therapy is a novel and highly efficacious immunotherapeutic modality, that generated hitherto unseen remission rates in relapsed/refractory B-cell malignancies and multiple myeloma, and that could be an attractive option for *TP53*-mutant AML/MDS as well (Berdeja et al, 2021; Davila et al, 2014; Kochenderfer et al, 2015; Lee et al, 2015a; Maude et al, 2014; Munshi et al, 2021; Neelapu et al, 2017; Schuster et al, 2019; Turtle et al, 2016a). A fundamental prerequisite for the development of an effective CAR T-cell therapy in AML/MDS would be determining the most suitable AML/MDS-associated cell

[1]Department of Medical Oncology and Hematology, University of Zurich and University Hospital Zurich, Zurich, Switzerland. [2]Department of Biosystems Science and Engineering, ETH Zurich, Basel, Switzerland. [3]Institute of Biomedicine and Medicity Research Laboratories, University of Turku, Turku, Finland. [4]Department of Medical Oncology, Dana-Farber Cancer Institute, Boston, MA, USA. [5]These authors contributed equally: Jan Mueller, Roman R Schimmer. [6]These authors contributed equally: Markus G Manz, Steffen Boettcher. ✉E-mail: steffen.boettcher@usz.ch

surface target antigen (Borot et al, 2019; Haubner et al, 2019; Perna et al, 2017). Current CAR T-cell therapeutic efforts in AML/MDS are, therefore, primarily devoted to testing such target antigens including the surface antigens CD33, CD117, CD123, and CD371 (Arai et al, 2018; Laborda et al, 2017; Mardiana and Gill, 2020; Myburgh et al, 2020; Tashiro et al, 2017; Wang et al, 2015). Equally important for therapeutic efficacy but much less well-studied are intrinsic and acquired resistance mechanism to CAR T-cell-mediated killing. While target antigen loss is a well-known acquired resistance mechanism observed in B-cell lymphoma and B-lymphoblastic leukemia (Fraietta et al, 2018; Shah and Fry, 2019; Sotillo et al, 2015), cancer cell-intrinsic determinants of the efficacy of CAR T-cell therapies are less well understood, particularly in AML (Singh et al, 2020). Loss of TP53 in target cells has been shown to confer resistance to cytotoxic T-cells and NK-cells via transcriptional and non-transcriptional mechanisms but how this might affect CAR T-cell-mediated killing is largely unknown (Ben Safta et al, 2015; Chollat-Namy et al, 2019; Thiery et al, 2005; Thiery et al, 2015). Furthermore, TP53 mutations in AML/MDS induce an immunosuppressive phenotype in the immune microenvironment with a potential for negatively impacting CAR T-cell efficacy (Sallman et al, 2020). Recently, in the clinical setting of diffuse large B-cell lymphoma, TP53 deficiency has been shown to be a negative prognostic marker of CD19-directed CAR T-cell therapy (Shouval et al, 2022), whereas another recent study could not replicate this finding (Jain et al, 2022). It is, therefore, currently unclear, whether the TP53 mutational status might impact the efficacy of CAR T-cells in hematological malignancies.

In light of the above-mentioned, we addressed the biologically and clinically highly relevant question as to whether, and, if so, how TP53 deficiency in AML cells might confer resistance to CAR T-cell therapy. To this end, we took advantage of a recently developed, CRISPR/Cas9-engineered MOLM13-TP53 isogenic human AML cell line model (Boettcher et al, 2019) as well as a newly generated isogenic AML cell line model (MV4-11), and tested the efficacy of CAR T-cells targeting common (CD33, CD117, CD123) AML cell surface antigens using flow cytometry-based in vitro killing assays, live-cell imaging, transcriptional profiling (mRNA-seq), CRISPR/Cas9-based gene editing in CAR T-cells, and a xenogeneic mouse model of AML.

## Results

### TP53-mutant AML cells are resistant to CAR T-cell-mediated killing in vitro

In order to test whether the TP53 mutational status of AML blasts might impact the efficacy of CAR T-cell therapies, we took advantage of a recently developed isogenic human MOLM13 AML cell line model that was CRISPR/Cas9-engineered to express an allelic series of the six most common TP53 missense mutations, null as well as wild-type alleles (Boettcher et al, 2019). Second-generation CAR T-cells with the 4-1BB costimulatory domain and the CD3ζ activating domain were generated by lentiviral transduction of T-cells from at least three different healthy donors (Fig. EV1A). The single-chain variable fragment (scFv) portion of the CAR was directed against three different target antigens present on normal myeloid precursors and most AML: CD33, CD123, and

CD117 (Figs. 1A and EV1A). CAR T-cells were expanded and purified by MACS to yield high purity (>95% by flow cytometry) products (Fig. EV1B). MOLM13-TP53 isogenic cells showed indistinguishable surface antigen expression and growth kinetics in the unperturbed state (Figs. 1B and EV1C).

When co-incubated with anti-CD33 CAR T-cells at various effector-to-target (E:T) ratios (Fig. 1A), we consistently observed significantly less killing of MOLM13-TP53 AML cells with knock-out (MOLM13-$TP53^{-/-}$) or missense mutant (MOLM13-$TP53^{missense/-}$) alleles as compared to MOLM13-TP53 AML cells with wild-type TP53 (MOLM13-$TP53^{+/+}$) at early (d1) or later (d6) time points (Fig. 1C–F). As a result, significantly more TP53-deficient MOLM13 cells survived the CAR T-cell attack and eventually outgrew the CAR T-cells in the co-culture (Fig. 1C,E). This held true for all the investigated target antigens (CD33, CD123) expressed on MOLM13-TP53 isogenic cell lines and over several log2 fold ranges of E:T ratios (Fig. 1D,F,G). CAR T-cells targeting CD117, which is not expressed on MOLM13-TP53 AML cells, did not show killing in any of the MOLM13-TP53 cell lines, irrespective of TP53 status (Fig. 1H). Co-incubation with untransduced T-cell controls did not lead to relevant AML cell killing irrespective of the TP53 genotype (Fig. 1C,E). For all subsequent experiments, we used CAR T-cells targeting CD33 and CD123 that are strongly expressed on MOLM13-TP53 isogenic AML cells (Fig. 1B). Given that we did not observe any differences between MOLM13-$TP53^{-/-}$ and MOLM13-$TP53^{missense/-}$ target cells with both CD33- and CD123-targeted CAR T-cells (Fig. 1F,G), underscoring that TP53 missense and null mutations are largely functionally equivalent (Boettcher et al, 2019), we used MOLM13-$TP53^{-/-}$ cells in subsequent experiments.

Co-incubation of CAR T-cells with MOLM13-$TP53^{+/+}$ and MOLM13-$TP53^{-/-}$ together in a competitive manner revealed the preferential killing of MOLM13-$TP53^{+/+}$ over MOLM13-$TP53^{-/-}$ AML cells and continuous outgrowth of MOLM13-$TP53^{-/-}$ AML cells in the co-cultures, irrespective of E:T or initial MOLM13-$TP53^{+/+}$/MOLM13-$TP53^{-/-}$ ratios (Fig. EV1D–G). Finally, we could independently validate the reduced killing of TP53-deficient isogenic cells in a CRISPR/Cas9-engineered TP53 isogenic MV4-11 AML cell line with TP53 wild-type or knockout alleles co-incubated with CAR T-cells targeting either CD33 or CD123, and across several log2 fold E:T ratios. (Fig. 1I).

Altogether, we revealed that TP53 deficiency in AML cells confers resistance against CAR T-cell-mediated killing in vitro.

### CAR T-cells engaging MOLM13-$TP53^{-/-}$ AML cells proliferate less, exhibit sustained activation marker expression, an exhausted immunophenotype, and enhanced trogocytosis

To elucidate the underlying mechanisms for decreased killing efficacy of CAR T-cells attacking MOLM13-$TP53^{-/-}$ cells, we first assessed CAR T-cell proliferation during 6 days of co-incubation. While CAR T-cells expanded 1–1.5 log-fold in the presence of MOLM13-$TP53^{+/+}$ cells, CAR T-cells co-incubated with MOLM13-$TP53^{-/-}$ or MOLM13-$TP53^{missense/-}$ cells showed a significantly lower expansion (Figs. 2A and EV1H). Next, we measured CAR T-cell activation as determined by the surface expression of the early T-cell activation marker CD25. The fraction of CAR T-cells with sustained CD25 expression at day 6 of co-incubation was

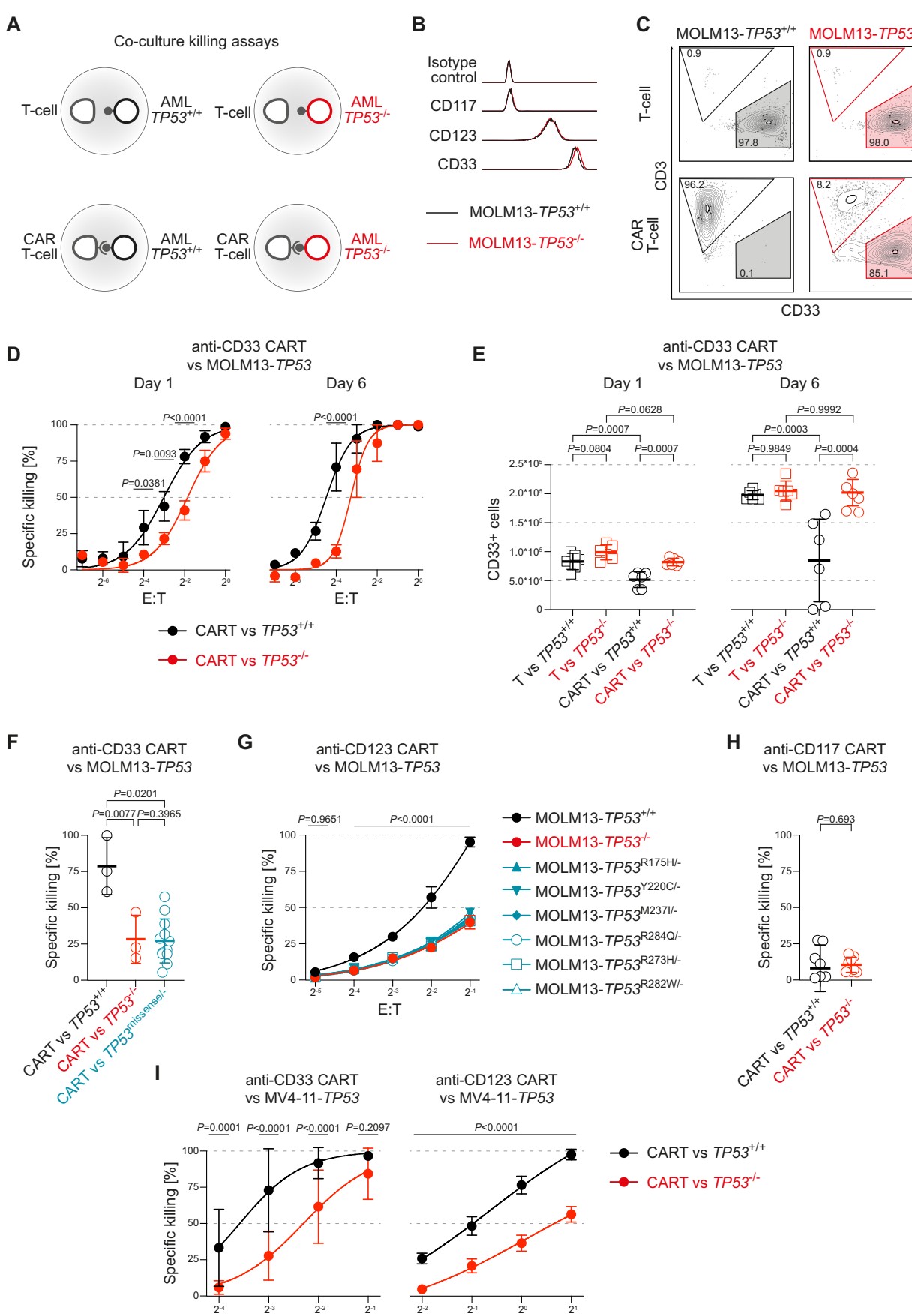

◄

**Figure 1. *TP53*-mutant AML cells are resistant to CAR T-cell-mediated killing in vitro.**

(A) Scheme of in vitro killing assays with untransduced T-cell controls, CAR T-cells, and isogenic MOLM13-*TP53* AML cells with different *TP53* status. (B) Representative expression of CD117 (negative), CD123 (int), CD33 (high) on isogenic MOLM13-*TP53* AML cells relative to isotype control. (C) Representative FACS plots of in vitro killing assays of anti-CD33 CAR T-cells with isogenic MOLM13-*TP53* AML cells with wild-type (MOLM13-*TP53*$^{+/+}$) or null (MOLM13-*TP53*$^{-/-}$) *TP53* status. Percentages of parental populations are shown. (D) Specific CAR T-cell-mediated killing of MOLM13-*TP53*$^{+/+}$ AML cells (black) or MOLM13-*TP53*$^{-/-}$ AML cells (red) at the indicated E:T ratios and incubation days. Pooled results from three different healthy T-cell donors are shown (biological replicates per healthy donor, $n = 3$; two technical replicates for each biological replicate; symbols represent averages of all replicates; error bars indicate SD; two-way ANOVA). (E) Absolute CD33+ cell numbers designating MOLM13-*TP53* AML cells (black: *TP53*$^{+/+}$, and red: *TP53*$^{-/-}$) in co-incubation assays with untransduced T-cells and anti-CD33 CAR T-cells at an E:T of 1:16 on days 1 and 6 (biological replicates, $n = 3$; two technical replicates for each biological replicate; symbols represent individual replicates; thickened line represents mean and error bars indicate SD; two-way ANOVA). (F) Specific anti-CD33 CAR T-cell-mediated killing of MOLM13-*TP53*$^{+/+}$ (black), MOLM13-*TP53*$^{-/-}$ (red) or MOLM13-*TP53*$^{missense/-}$ (blue) AML cells on day 6 of co-incubation (biological replicates, $n = 2$ for MOLM13-*TP53*$^{+/+}$, MOLM13-*TP53*$^{-/-}$, and for each of the six indicated MOLM13-*TP53*$^{missense/-}$ conditions; two technical replicates for each biological replicate; symbols represent individual replicates; thickened line represents mean and error bars indicate SD; two-way ANOVA). (G) Specific killing of anti-CD123 CAR T-cells against MOLM13-*TP53* AML cells at various E:T ratios (biological replicates, $n = 3$–4; symbols represent means; error bars indicate SD; two-way ANOVA). (H) Specific CAR T-cell-mediated killing of MOLM13-*TP53*$^{+/+}$ AML cells (black) and MOLM13-*TP53*$^{-/-}$ AML cells (red) with a CAR targeting CD117 (D79), (biological replicates, $n = 3$; two technical replicates for each biological replicate; symbols represent individual replicates; thickened line represents mean and error bars indicate SD; unpaired Student's *t* test). (I) Left panel: Specific killing of anti-CD33 CAR T-cell against MV4-11-*TP53* AML cells at various E:T ratios (biological replicates, $n = 3$; symbols represent means; error bars indicate SD; two-way ANOVA). Right panel: Specific killing of anti-CD123 CAR T-cells against MV4-11-*TP53* AML cells at various E:T ratios (biological replicates, $n = 3$; symbols represent means; error bars indicate SD; two-way ANOVA). Source data are available online for this figure.

significantly higher in CAR T-cells engaging MOLM13-*TP53*$^{-/-}$ AML cells than those engaging MOLM13-*TP53*$^{+/+}$ AML cells (Fig. 2B,C; Appendix Fig. S1).

As sustained stimulatory signaling via the CAR is associated with CAR T-cell exhaustion, we next analyzed the expression of canonical T-cell markers that are associated with insufficient clearance of antigen-expressing cells, i.e., PD-1 (Programmed cell death protein 1), LAG3 (Lymphocyte activating 3), and TIM3 (T-cell immunoglobulin and mucin-domain containing-3). We observed *TP53*-dependent differences in exhaustion profiles: CAR T-cells co-incubated with MOLM13-*TP53*$^{-/-}$ AML cells exhibited a more exhausted T-cell phenotype than CAR T-cells co-incubated with MOLM13-*TP53*$^{+/+}$ AML cells as demonstrated by a higher percentage of CAR T-cells being double- or triple-positive for the above exhaustion markers (Fig. 2D,E; Appendix Fig. S2). Untransduced T-cells did not show upregulated surface expression of exhaustion markers upon co-incubation with either AML cell line (Fig. 2D,E; Appendix Fig. S2).

Trogocytosis is the active process, described in CAR T-cells, of acquiring target antigens while in contact with their target cells, followed by displaying them incorporated into their own membrane surface (Hamieh et al, 2019). We found a significantly higher percentage of acquired CD33 expression on CAR T-cells co-incubated with MOLM13-*TP53*$^{-/-}$ AML cells than on CAR T-cells co-incubated with MOLM13-*TP53*$^{+/+}$ AML cells or the untransduced T-cells in control conditions, hinting at a longer and/or more intimate CAR T-cell:AML cell interaction in the context of *TP53* deficiency in AML cells (Fig. 2F,G; Appendix Fig. S3). Although trogocytosis has been recognized as a means of acquired resistance to CAR T-cell therapies (Hamieh et al, 2019) via reversible antigen loss on tumor cells, we did not observe a decrease in target antigen density on MOLM13-*TP53*$^{-/-}$ AML cells (Fig. EV1I; Appendix Fig. S4A). We did neither observe evidence of antigen-negative escape by outgrowth of CD33$^-$ MOLM13-*TP53* isogenic AML cells, nor differences in surface expression of the canonical immune checkpoint marker PD-L1 (Programmed cell death 1 ligand 1) at steady state or upon co-incubation with CAR T-cells (Fig. EV1J; Appendix Fig. S4A,B).

In summary, the inability of CAR T-cells to eradicate MOLM13-*TP53*$^{-/-}$ AML cells in vitro likely involves sustained activating CAR signaling, exacerbated CAR T-cell exhaustion, enhanced trogocytosis, and decreased proliferation.

## Live-cell imaging shows a longer duration of interaction between CAR T-cells and MOLM13-*TP53*$^{-/-}$ AML cells

Increased trogocytosis in CAR T-cells attacking MOLM13-*TP53*$^{-/-}$ cells suggests a longer duration of engagement between CAR T-cells and *TP53*-deficient AML cells. To further investigate CAR T-cell:AML cell interaction dynamics, we employed fluorescence live-cell microscopy. CAR T-cells were co-incubated with isogenic MOLM13-*TP53* AML cells with either *TP53* wild-type or *TP53* null status at an E:T ratio of 1:1 on imaging slides containing microgrids to avoid cell clustering and allow for tracking of individual killing events (Fig. 2H; Movies EV1 and EV2). The time from engagement between CAR T-cells and AML cells, and subsequent AML cell lysis, as judged by propidium iodide (PI) influx, was significantly longer, i.e., increased from roughly 10 h to ~16 h, for CAR T-cells co-incubated with MOLM13-*TP53*$^{-/-}$ as compared to MOLM13-*TP53*$^{+/+}$ AML cells (Fig. 2H,I; Movies EV1 and EV2). Under these specific experimental conditions of a high E:T ratio with cells confined to microgrids, the *TP53* genotype-specific difference between MOLM13-*TP53*$^{-/-}$ and MOLM13-*TP53*$^{+/+}$ AML cells killed by CAR T-cells was less pronounced than in the regular co-culture assays. However, there were still significantly fewer killing events upon CAR T-cell co-incubation with MOLM13-*TP53*$^{-/-}$ than with MOLM13-*TP53*$^{+/+}$ AML cells (Fig. 2J).

Taken together, these data suggest that the delayed CAR T-cell-mediated killing of MOLM13-*TP53*$^{-/-}$ AML cells, and therefore prolonged antigen exposure to CAR T-cells likely contributes to the observed sustained activation, increased exhaustion, and consequently, lower therapeutic efficacy against *TP53*-mutant AML cell lines in vitro.

## MOLM13-*TP53*$^{-/-}$ AML cells are resistant to CAR T-cells in a therapeutic xenograft in vivo mouse model

Having established that loss of *TP53* in AML cells promotes increased resistance to CAR T-cell therapies in vitro, we next employed a therapeutic AML mouse model to test the in vivo relevance of our findings. Isogenic MOLM13-*TP53* AML cells with either *TP53*$^{-/-}$ or *TP53*$^{+/+}$ status were transduced with a luciferase lentiviral construct (Myburgh et al, 2020), allowing us to track cells in vivo by bioluminescence. One hundred thousand

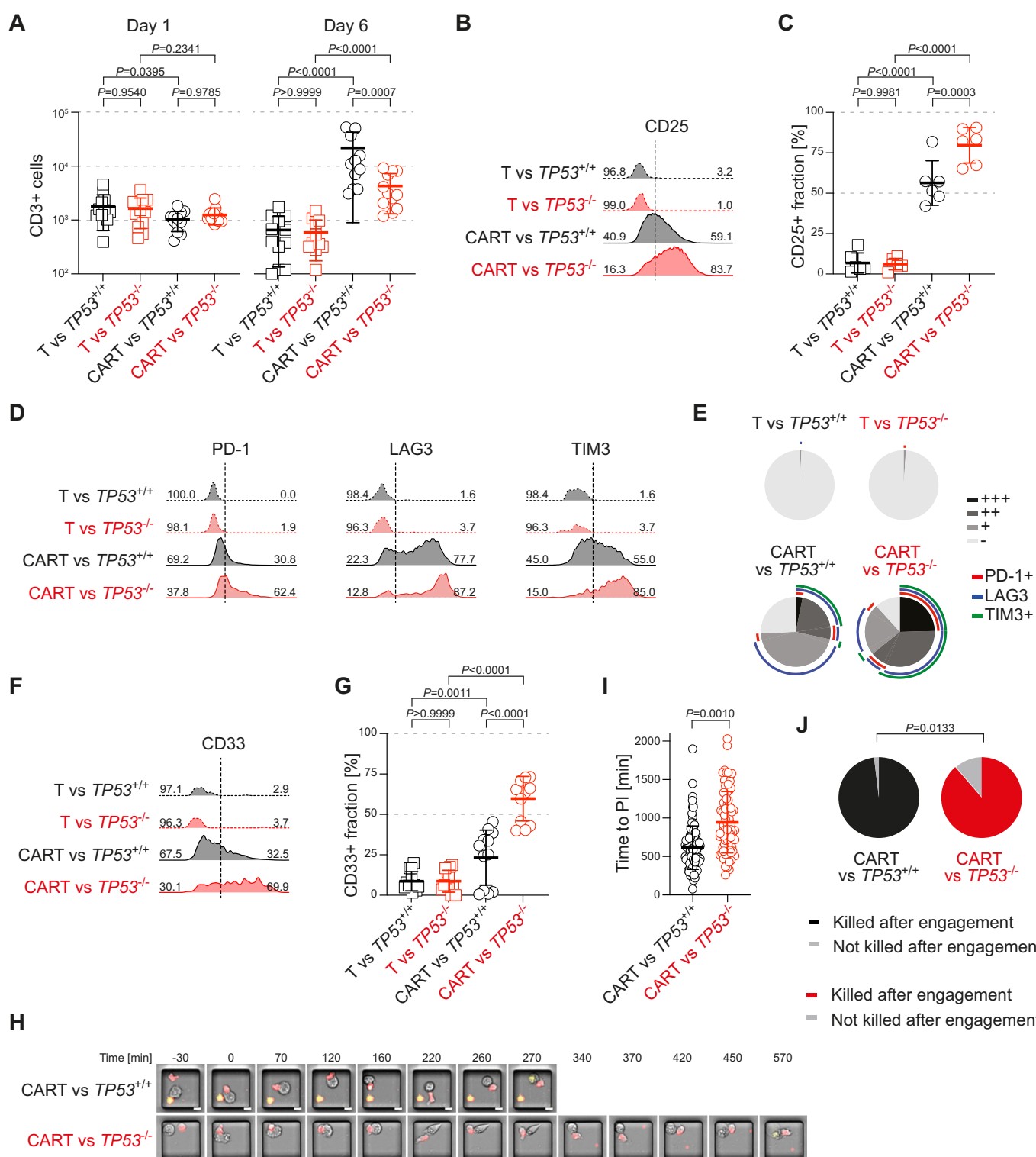

luciferase-expressing MOLM13-$TP53^{+/+}$ or MOLM13-$TP53^{-/-}$ AML cells were injected into the tail vein of sublethally irradiated immunodeficient NOD-scid IL2Rγ$^{null}$ (NSG) mice. After 7 days, $2\text{–}8 \times 10^6$ anti-CD33 CAR T-cells or T-cell controls were injected via the tail vein (Fig. 3A). Mice injected with control T-cells showed rapid AML progression as determined by bioluminescence imaging

and had to be sacrificed within 3 weeks because of symptomatic leukemic disease irrespective of the $TP53$ genotype of the injected MOLM13 cells (Figs. 3B–D and EV2). Mice injected with MOLM13-$TP53^{-/-}$ AML cells and treated with anti-CD33 CAR T-cells started to succumb to leukemic disease within 3 weeks as well. By contrast, mice injected with MOLM13-$TP53^{+/+}$ AML cells

**Figure 2. CAR T-cells engaging MOLM13-*TP53*^−/− AML cells proliferate less, exhibit sustained activation marker expression, an exhausted immunophenotype, enhanced trogocytosis, and a longer duration of interaction between CAR T-cells and MOLM13-*TP53*^−/− AML cells.**

(A) Absolute CD3+ cell numbers for untransduced T-cell controls and anti-CD33 CAR T-cells co-incubated with MOLM13-*TP53* AML cells (black: *TP53*^+/+, and red: *TP53*^−/−) at an E:T of 1:16 on days 1 and 6 (biological replicates, n = 4; 2–3 technical replicates for each biological replicate; symbols represent individual replicates; thickened line represents mean and error bars indicate SD; two-way ANOVA). (B) Representative FACS histograms with according percentages of CD25 positive CAR T-cells and T-cell controls co-incubated with isogenic MOLM13-*TP53* AML cells with wild-type (MOLM13-*TP53*^+/+, in black) or null (MOLM13-*TP53*^−/−, in red) *TP53* status. (C) Quantitative CD25 expression on CAR T-cells and T-cell controls co-incubated for 6 days with isogenic MOLM13-*TP53* AML cells with wild-type (MOLM13-*TP53*^+/+, in black) or null (MOLM13-*TP53*^−/−, in red) *TP53* status, (biological replicates, n = 3; two technical replicates per biological replicate; symbols represent individual replicates; thickened lines indicate mean and error bars indicate SD; two-way ANOVA). (D) Representative FACS histograms with according percentages of PD-1, LAG3, and TIM3 positive CAR T-cells and T-cell controls co-incubated with isogenic MOLM13-*TP53* AML cells with wild-type (MOLM13-*TP53*^+/+, in black) or null (MOLM13-*TP53*^−/−, in red) *TP53* status on day 6 of co-incubation. (E) Summary of exhaustion markers of CAR T-cells and T-cell controls co-incubated with isogenic MOLM13-*TP53* AML cells with wild-type (MOLM13-*TP53*^+/+) or null (MOLM13-*TP53*^−/−) *TP53* status (biological replicates, n = 3; +, single exhaustion marker positive; ++, double exhaustion marker positive; +++, triple exhaustion marker positive; −, negative for exhaustion markers). (F) Representative FACS histograms of CD33 expression on CAR T-cells and T-cell controls co-incubated with isogenic MOLM13-*TP53* AML cells with wild-type (MOLM13-*TP53*^+/+, in black) or null (MOLM13-*TP53*^−/−, in red) *TP53* status. (G) Quantification of trogocytosis of CAR T-cells and T-cell controls co-incubated with isogenic MOLM13-*TP53* AML cells with wild-type (MOLM13-*TP53*^+/+, in black) or null (MOLM13-*TP53*^−/−, in red) *TP53* status on day 6 of co-incubation, (biological replicates, n = 4; 2–3 technical replicates per biological replicate; symbols represent individual replicates; thickened lines indicate mean and error bars indicate SD; two-way ANOVA). (H) Representative stills of fluorescence live-cell time-lapse imaging data of anti-CD33 CAR T-cell engaging MOLM13-TP53^+/+ AML or MOLM13-TP53^−/− AML cells (scale bars, 10 μm). (I) Summary data showing time to propidium iodide influx for CAR T-cells engaging MOLM13-*TP53*^−/− and MOLM13-*TP53*^+/+ AML cells (biological replicates, n = 3; >17 technical replicates per biological replicate; symbols indicate individual replicates; thickened lines indicate mean and error bars indicate SD; unpaired Student's *t* test). (J) Fraction of killing events as judged by propidium iodide influx after engagement of CAR T-cells to MOLM13-*TP53* isogenic cell lines (Fisher's exact test). Source data are available online for this figure.

and treated with anti-CD33 CAR T-cells showed decreased leukemic burden, and significantly prolonged survival for up to 10 weeks (Figs. 3B–D and EV2).

Altogether, these data confirm our above-described results from in vitro co-culture experiments corroborating our finding that the *TP53* mutational status is a critical tumor-intrinsic genetic determinant of CAR T-cell therapeutic efficacy in AML in vitro and in vivo.

## Gene expression analysis of CAR T-cells and isogenic MOLM13-*TP53* AML cells reveals differentially expressed *TP53*-dependent genes and pathways

To investigate the underlying molecular mechanisms of the observed increased resistance of *TP53*-deficient AML cells to CAR T-cell attack, we performed gene expression analyses of resting anti-CD33 CAR T-cells as well as isogenic MOLM13-*TP53*^−/− and MOLM13-*TP53*^+/+ AML cells. Furthermore, we set up co-culture assays between CAR T-cells and MOLM13-*TP53*^−/− or MOLM13-*TP53*^+/+ AML cells, respectively, and FACS-sorted all CAR T-cell or AML cell populations after 6 days of co-incubation (Figs. 4A and EV3A–E). We termed the resulting seven distinct cell populations CART(resting), MOLM13-*TP53*^+/+(resting), MOLM13-*TP53*^−/−(resting), CART(MOLM13-*TP53*^+/+), CART(MOLM13-*TP53*^−/−), MOLM13-*TP53*^+/+(CART), and MOLM13-*TP53*^−/−(CART) (Fig. 4A). After bulk mRNA sequencing (mRNA-seq) of the described seven cell populations, principal component analysis (PCA) showed clustering of triplicates of the respective populations, with the highest variation separating CAR T-cells from MOLM13-*TP53* AML cells (Figs. 4B and EV3F).

First, we focused our analysis on differentially regulated genes in isogenic MOLM13-*TP53* cells (Figs. 4C–E and EV3G; Appendix Fig. S5A). We found genes involved in the mevalonate pathway to be the most differentially expressed gene set upregulated in MOLM13-*TP53*^−/− attacked by CAR T-cells, i.e., MOLM13-*TP53*^−/−(CART) as compared to MOLM13-*TP53*^+/+ attacked by CAR T-cells, i.e., MOLM13-*TP53*^+/+(CART) (Figs. 4C–E and EV3G). We validated our findings by measuring the expression of the canonical mevalonate pathway genes *HMGCR* and *HMGCS1* via RT-qPCR in the setting of anti-CD33 as well as anti-CD123

CAR T-cell attack in isogenic MOLM13-*TP53* and MV4-11-*TP53* AML cell lines (Fig. EV3H).

Second, we analyzed differential gene expression in resting and co-incubated T-cell populations (Figs. 4F and EV3I; Appendix Fig. S5B). Among the most differentially expressed genes between CAR T-cells attacking MOLM13-*TP53*^−/− AML cells, i.e., CART(MOLM13-*TP53*^−/−) and CAR T-cells attacking MOLM13-*TP53*^+/+ AML cells, i.e., CART(MOLM13-*TP53*^+/+) we identified two known master regulators of T-cell fate and downstream effectors of the Wnt signaling pathway *TCF7* (Transcription factor 7, also known as T-cell-specific transcription factor 1, *TCF-1*), and *EOMES* (Eomesodermin) (Fig. 4F). The differential expression of *TCF7* and *EOMES* was confirmed by RT-qPCR in anti-CD33 and anti-CD123 CAR T-cells engaging isogenic MOLM13-*TP53* as well as isogenic MV4-11-*TP53* AML cell lines (Fig. EV3J).

These findings suggest that during engagement between CAR T-cells and AML cells both CAR T-cell-intrinsic as well as AML cell-intrinsic pathways are differentially activated or repressed, which in turn, may subsequently mediate CAR T-cell efficacy or lack thereof, respectively. Notably, the precise nature of these responses is dependent on the *TP53* mutational status of the AML cells. This might provide an opportunity for a therapeutic intervention to enhance the efficacy of CAR T-cells against *TP53*-mutant AML.

## Targeting the mevalonate or the Wnt pathway fully rescues CAR T-cell killing of *TP53*-deficient AML cells

In order to test whether the mevalonate and/or Wnt pathways, prominent members of which we found to be differentially expressed in a *TP53*-dependent manner, are involved in determining the efficacy of CAR T-cell-mediated killing of *TP53*-deficient AML cells, we sought to target these pathways pharmacologically in co-culture killing assays in vitro (Fig. 5A).

In terms of the mevalonate pathway in *TP53*-mutant AML cells, we investigated the effect of the competitive HMG-CoA reductase (*HMGCR*; 3-hydroxy-3-methylglutaryl coenzyme A reductase) inhibitor simvastatin, which blocks the rate-limiting step of the mevalonate pathway. Furthermore, we supplemented the

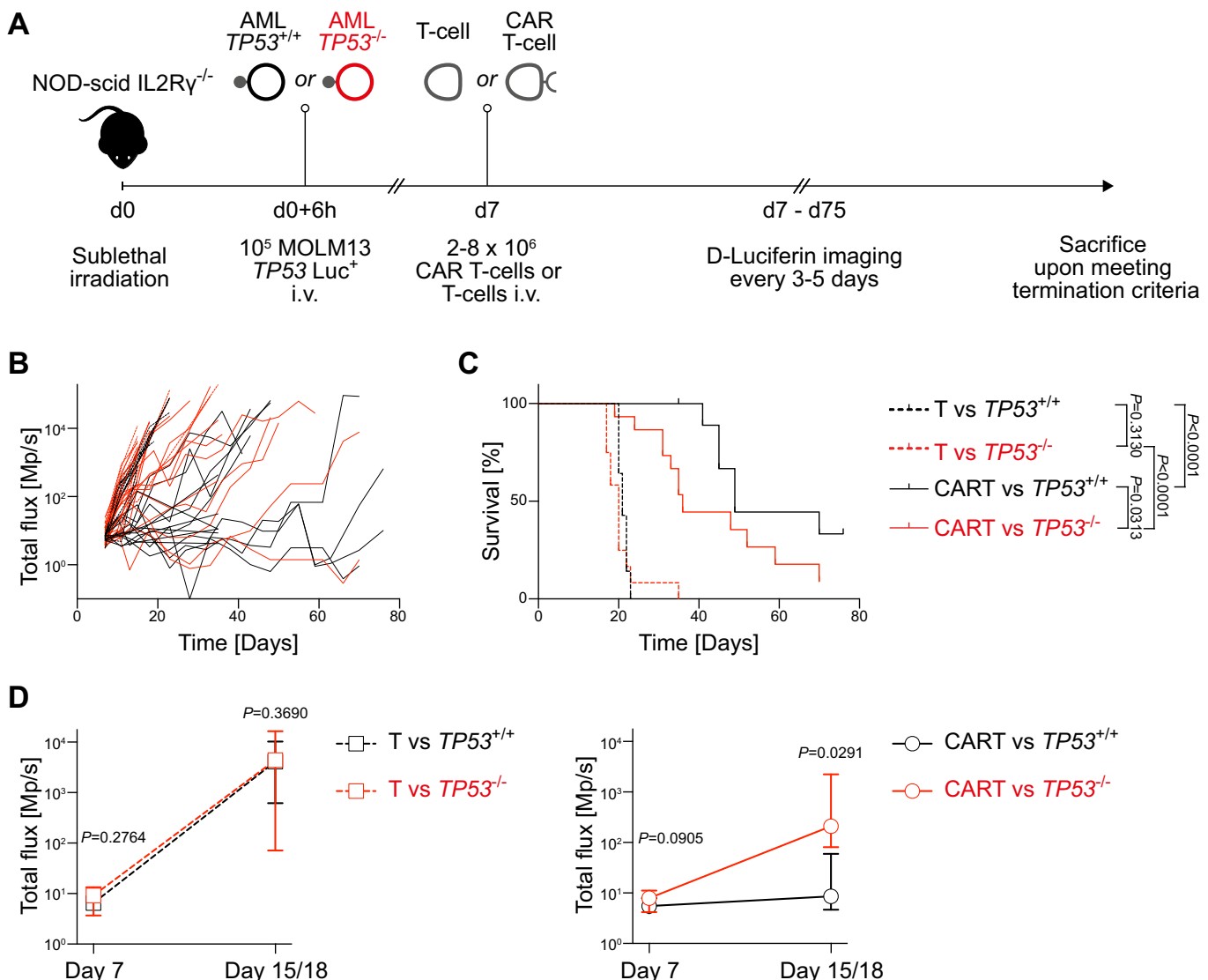

**Figure 3. MOLM13-*TP53*⁻/⁻ AML cells are resistant to CAR T-cells in a therapeutic xenograft in vivo mouse model.**

(A) Experimental outline depicting the overall workflow of the xenograft mouse model. (B) Summary of bioluminescence signals longitudinally measured in mice engrafted with luciferase-expressing MOLM13-*TP53*⁺/⁺ or MOLM13-*TP53*⁻/⁻ AML cells and treated with anti-CD33 CAR T-cells or untransduced T-cell controls from three pooled independent experiments (n = 53 mice in total and n = 3 healthy donors, T-cells vs. MOLM13-*TP53*⁺/⁺Luc⁺=12, T-cells vs. MOLM13-*TP53*⁻/⁻Luc⁺=11, CAR T-cells vs. MOLM13-*TP53*⁺/⁺Luc⁺=15, CAR T-cells vs. MOLM13-*TP53*⁻/⁻Luc⁺=15). (C) Kaplan–Meier survival curve of mice engrafted with luciferase-expressing MOLM13-*TP53*⁺/⁺ or MOLM13-*TP53*⁻/⁻ AML cells and treated with anti-CD33 CAR T-cells or untransduced T-cell control from three pooled independent experiments. (n = 53 mice in total and n = 3 healthy donors, T-cells vs. MOLM13-*TP53*⁺/⁺Luc⁺=12, T-cells vs. MOLM13-*TP53*⁻/⁻Luc⁺=11, CAR T-cells vs. MOLM13-*TP53*⁺/⁺Luc⁺=15, CAR T-cells vs. MOLM13-*TP53*⁻/⁻Luc⁺=15; symbols represent mean; error bars indicate SD; Log-rank Mantel–Cox test). (D) Pooled bioluminescence signals on day 7 (before CAR T-cell or untransduced T-cell infusion) and on the last day before termination of any mice (day 15 or 18 for the different biological replicates). Bioluminescence signals of mice injected with T-cell controls are shown on the left and bioluminescence signals of mice injected with CAR T-cells are shown on the right. (n = 53 mice in total and n = 3 healthy donors, T-cells vs. MOLM13-*TP53*⁺/⁺Luc⁺=12, T-cells vs. MOLM13-*TP53*⁻/⁻Luc⁺=11, CAR T-cells vs. MOLM13-*TP53*⁺/⁺Luc⁺=15, CAR T-cells vs. MOLM13-*TP53*⁻/⁻Luc⁺=15; unpaired Student's *t* test). Source data are available online for this figure.

simvastatin-treated cultures with the HMGCR metabolite mevalonate to reverse the effect of simvastatin. Through a series of experiments measuring low-density lipoprotein receptor (LDLR) surface expression by FACS and the expression of the sterol-regulatory element (SRE) in an established luciferase reporter assay, we confirmed the expected on-target activity of simvastatin (Fig. EV4A–D; Appendix Fig. S5C). For all future experiments, we used 1 μM and 15 μM simvastatin for experiments with MOLM13 and

MV4-11 cells, respectively, as determined in dose-titration experiments (Fig. EV4A,B).

We first performed a series of in vitro co-incubation experiments: Adding simvastatin at the previously established dose to our killing assays completely rescued the sensitivity of MOLM13-*TP53*⁻/⁻ or MV4-11-*TP53*⁻/⁻ AML cells co-incubated with CAR T-cells over several log2 fold E:T range with CD33- as well as CD123-targeting CAR T-cells (Figs. 5B,C and EV4E,F). Of note, the addition of

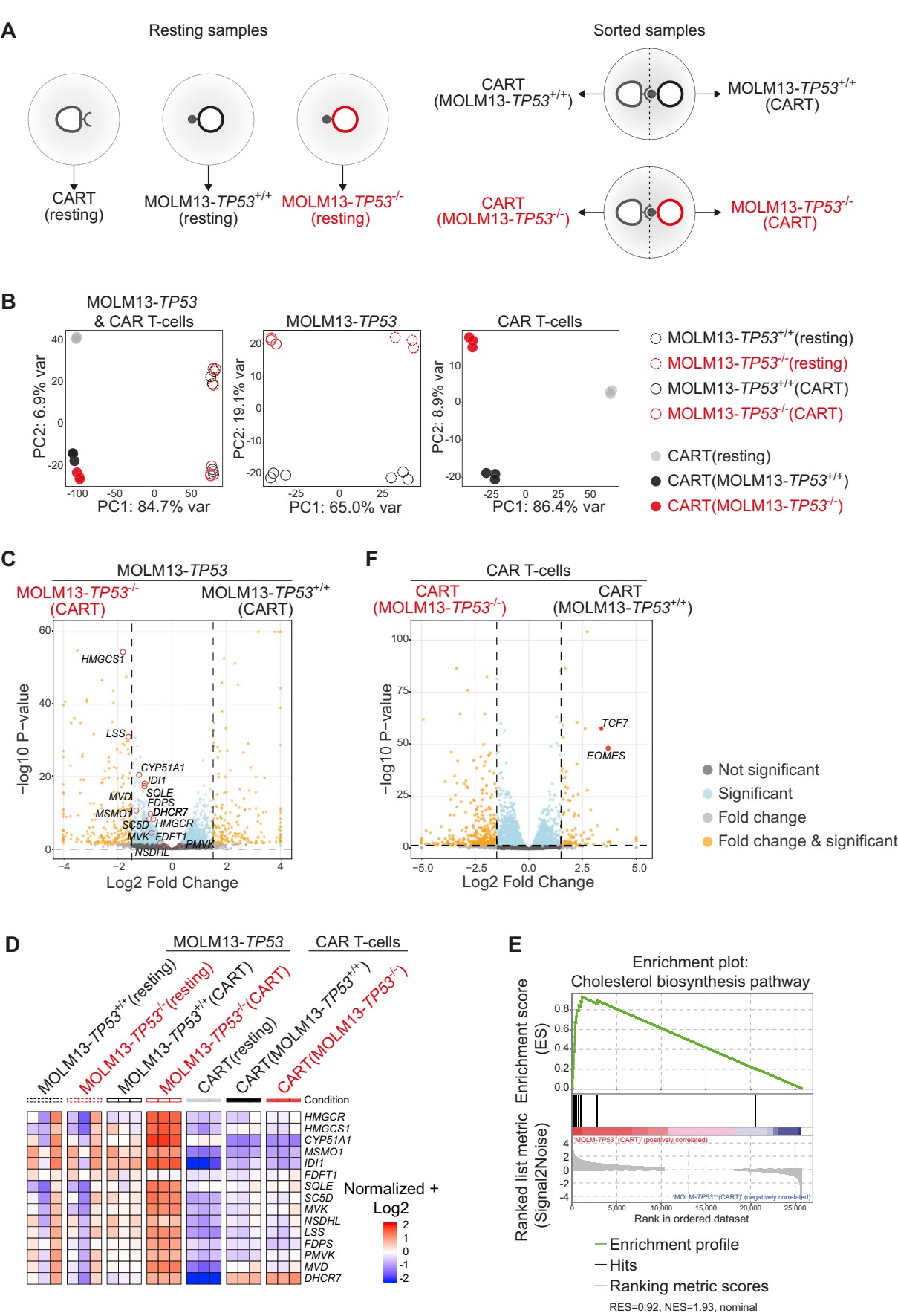

**Figure 4. Gene expression analysis of CAR T-cells and isogenic MOLM13-*TP53* AML cells reveals differentially expressed *TP53*-dependent genes and pathways.**

(A) Overview of the samples obtained for mRNA-seq. (B) Principal component analysis (PCA) of the seven distinct samples with each of the three technical replicates. (C) Volcano plot of differentially expressed genes between MOLM13-*TP53*$^{+/+}$(CART) and MOLM13-*TP53*$^{-/-}$(CART) (individual data points represent means of $n = 3$ replicates per sample). (D) Heatmap showing differential expression of genes involved in cholesterol biosynthesis identified in the GSEA ($n = 3$ replicates per sample). (E) Gene set enrichment analysis (GSEA) of mevalonate pathway genes in MOLM13-*TP53*$^{-/-}$(CART) vs MOLM13-*TP53*$^{+/+}$(CART), (ES = 0.92, NES = 1.93, nominal $P$ value = <0.000, FDR $q$-value = <0.000, Kolmogorov–Smirnov test). (F) Volcano plot of differentially expressed genes between CART(MOLM13-*TP53*$^{+/+}$) and CART(MOLM13-*TP53*$^{-/-}$) (individual data points represent means of $n = 3$ replicates per sample).

simvastatin to co-cultures of CAR T-cells and AML cells with wild-type *TP53* did not lead to increased killing of AML cells (Fig. 5B,C and EV4E,F). The addition of mevalonate to simvastatin-containing co-incubation assays completely restored the *TP53* deficiency-associated resistance of AML cells to CAR T-cell killing with no effect on killing of MOLM13-*TP53*$^{+/+}$ AML cells (Fig. 5B,C and EV4E).

Assaying the established hallmarks of CAR T-cell dysfunction in the context of *TP53*-mutant AML (Fig. 2), we observed a rescue of CAR T-cell expansion upon simvastatin addition (Fig. EV4G). CAR T-cells co-incubated with MOLM13-*TP53*$^{-/-}$ AML cells in the presence of simvastatin showed reduced exhaustion markers (PD-1, LAG3, TIM3) compared to the DMSO controls (Figs. 5D and EV5A), and fluorescence live-cell imaging revealed normalization of CAR T-cell:AML cell interaction times upon addition of simvastatin (Figs. 5E and EV5B).

In terms of the Wnt pathway in CAR T-cells attacking *TP53*-mutant AML cells, we sought to pharmacologically increase the activation of the canonical Wnt pathway in CAR T-cells. To this end, we added 0.5 μM of the GSK-3 inhibitor and thus activator of the canonical Wnt pathway, BIO-acetoxime to our killing assays (Fig. 5A). We confirmed on-target activity of BIO-acetoxime by a luciferase reporter system in T-cells (Fig. EV5C).

BIO-acetoxime treatment of the co-culture assays resulted in a complete rescue of the deficiency of CAR T-cells in killing MOLM13-*TP53*$^{-/-}$ AML cells over several log2-fold range of E:T ratios as well as a reversal of most of the CAR T-cell dysfunction features: We observed decreased exhaustion marker expression in CAR T-cells attacking MOLM-*TP53*$^{-/-}$ AML cells, normalized interaction times, and increased T-cell proliferation compared to control conditions (Figs. 5B–E, EV4E–G and EV5A,B). Additional killing assays using anti-CD33 and anti-CD123-targeting CAR T-cells together with isogenic MOLM13-*TP53* and MV4-11-*TP53* AML cells confirmed the complete reversal of the resistance of *TP53*-mutant AML cells against CAR T-cell killing (Fig. EV4E).

## Pretreatment with simvastatin but not with BIO-acetoxime rescues CAR T-cell killing of *TP53*-deficient AML cells

Our transcriptional profiling results suggest that the effect of the mevalonate and Wnt pathways in mediating resistance of *TP53*-mutant AML cells to CAR T-cells are restricted to *TP53*-mutant AML and CAR T-cells attacking *TP53*-mutant AML cells, respectively. In order to test this functionally, we performed a second set of co-incubation experiments, this time with simvastatin- or BIO-acetoxime-pretreated AML or CAR T-cells (Fig. 6A).

As a first step, we analyzed the kinetics of the on-target activity of both drugs at the established concentrations for several days after washout: we observed sustained upregulation of LDLR over the course of 6 days after removal of simvastatin in both analyzed AML

cell lines and genotypes (Figs. 6B and EV4D). By contrast, washout of BIO-acetoxime led to a rapid normalization of the *TCF*-reporter signal with indistinguishable levels by day 4 after drug removal (Figs. 6C and EV5C).

Consistent with the prolonged simvastatin effect as shown in the washout experiments, simvastatin pretreatment of AML cells before co-incubation with CAR T-cells led to a complete rescue of the specific killing of both *TP53*-mutant MOLM13 and MV4-11 AML cells (Fig. 6D,E). Simvastatin pretreatment of AML cell lines with wild-type *TP53*$^{+/+}$ or CAR T-cells did not show any effect on CAR T-cell efficacy, indicating that the effect is specific for *TP53*-deficient AML cells (Figs. 6D,E and EV5D). These results were consistently observed with CD33- and CD123-directed CAR T-cells (Figs. 6D,E and EV5D). In line with our prior washout experiments and the rescued killing rate, we observed sustained LDLR upregulation in simvastatin pretreated AML cells, equivalent to AML cells incubated in the continuous presence of simvastatin, upon 6 days of co-incubation in our killing assays (Fig. EV5E). This observation held true for MOLM13-*TP53* and MV4-11-*TP53* cells and with CD33- as well as CD123-targeting CAR T-cells (Fig. EV5E).

However, BIO-acetoxime pretreatment of both AML cells and CAR T-cells failed to rescue the specific killing rate of *TP53*-mutant MOLM13 or MV4-11 AML cells by CAR T-cells (Fig. 6A,D,E and EV5D)—consistent with above washout experiments, which showed rapid normalization of Wnt signaling upon drug removal (Figs. 6C and EV5C).

Collectively, we here show that rational drug targeting of specific resistance mechanisms—i.e., the mevalonate pathway in *TP53*-mutant AML cells or the Wnt pathway in CAR T-cells engaging them—can completely rescue *TP53*-dependent resistance of AML cells to CAR T-cell killing in vitro.

## CRISPR/Cas9 knockout of Regnase-1 results in Wnt pathway upregulation and rescue of *TP53* deficiency-associated CAR T-cell resistance

Despite these encouraging results suggesting that pharmacological co-interventions with simvastatin and/or BIO-acetoxime may improve the efficacy of CAR T-cells against *TP53*-mutant AML, toxicity associated with such high doses of simvastatin as well as the short-lived effect of BIO-acetoxime would raise concerns over their clinical applicability. We therefore tested a genetic strategy to achieve sustained Wnt pathway activation in CAR T-cells.

To this end, we targeted Regnase-1 (also termed MCPIP1 and encoded by *ZC3H12A*), a rapid-response ribonuclease primarily involved in immune regulation, with *TCF7* as one of its primary targets (Matsushita et al, 2009; Uehata et al, 2013; Wei et al, 2019). Genetic manipulation of Regnase-1 has been investigated as a means of increasing *TCF7* signaling for improved antitumor

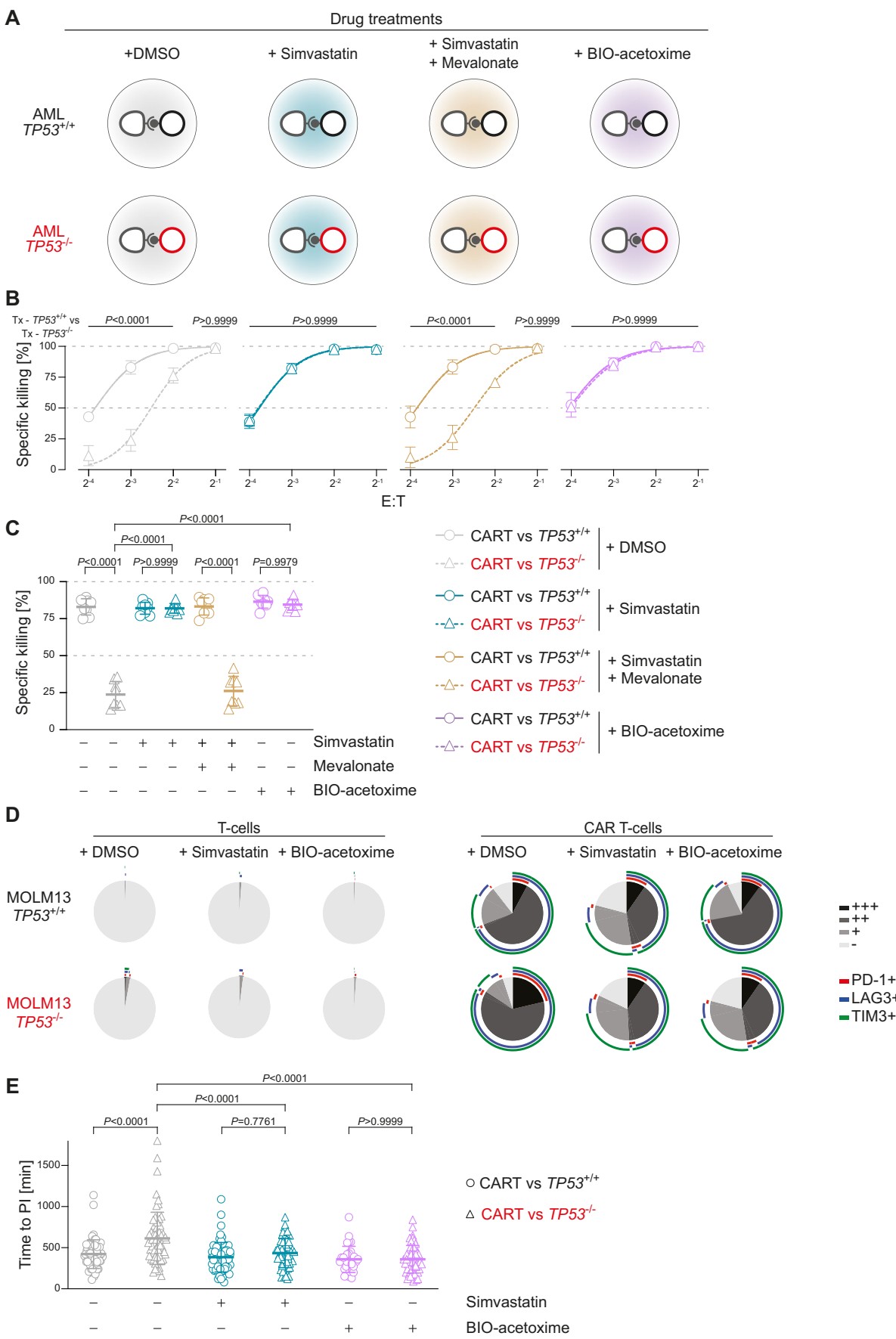

**Figure 5.  Targeting the mevalonate or the Wnt pathways fully rescues CAR T-cell killing of *TP53*-deficient AML cells.**

(A) Experimental outline showing the various co-incubation settings of CAR T-cells with *TP53*$^{+/+}$ or *TP53*$^{-/-}$ AML cells in the presence of DMSO, simvastatin, simvastatin+ mevalonate and BIO-acetoxime. (B) Summary data showing specific anti-CD33 CAR T-cell-mediated killing of MOLM13-*TP53*$^{+/+}$ or MOLM13-*TP53*$^{-/-}$ AML cells over various E:T ratios in the presence of DMSO, simvastatin 1 μM, simvastatin 1 μM + mevalonate 1 mM and BIO-acetoxime 0.5 μM, respectively. (biological replicates, $n = 3$; symbols represent mean; error bars indicate SD; two-way ANOVA). (C) Summary data showing results from (B) at an E:T of 1:8 (biological replicates, $n = 3$; two technical replicates per biological replicate; symbols represent individual replicates; error bars indicate SD; two-way ANOVA). (D) Summary of exhaustion markers of CAR T-cells and T-cell controls co-incubated with isogenic MOLM13-*TP53* AML cells with wild-type (MOLM13-*TP53*$^{+/+}$) or null (MOLM13-*TP53*$^{-/-}$) *TP53* status in the presence of DMSO, simvastatin 1 μM or BIO-acetoxime 0.5 μM (biological replicates, $n = 3$; +, single exhaustion marker positive; ++, double exhaustion marker positive; +++, triple exhaustion marker positive; −, negative for exhaustion markers). (E) Summary data showing time to propidium iodide influx for CAR T-cells engaging MOLM13-*TP53*$^{-/-}$ or MOLM13-*TP53*$^{+/+}$ AML cells in the presence of simvastatin 1 μM or BIO-acetoxime 0.5 μM (biological replicates, $n = 3$; symbols indicate individual technical replicates; error bars indicate SD; unpaired Student's *t* test). Source data are available online for this figure.

effector function (Mai et al, 2023; Wei et al, 2019; Zheng et al, 2021). To test whether Regnase-1 depletion in CAR T-cells could overcome the resistance of *TP53*-mutant AML cells against CAR T-cells, we knocked out Regnase-1 in CD33-directed as well as CD123-directed CAR T-cells using CRISPR/Cas9 genome editing (Fig. 6F). Using a previously published Regnase-1-targeting small guide RNA (sgRNA) (Mai et al, 2023), we generated CAR T-cells with high levels of Regnase-1 gene editing (>95% indels, Fig. EV5F). As expected, gene-edited Reg$^{-/-}$ CAR T-cells showed increased *TCF7* mRNA expression in comparison to untreated and CAR T-cells electroporated with non-targeting sgRNA (mock) (Fig. EV5G). Reg$^{-/-}$ CAR T-cells, co-incubated with isogenic MOLM13-*TP53* as well as MV4-11-*TP53* AML cells in our established killing assays, dramatically outperformed the untreated and mock CAR T-cell controls both against *TP53*$^{+/+}$ as well as *TP53*$^{-/-}$ AML cells over several E:T ratios and in both MOLM13- and MV4-11-*TP53* cells (Fig. 6G,H). The increased killing efficacy of Reg$^{-/-}$ CAR T-cells was even more pronounced in *TP53*$^{-/-}$ cells, leading to comparable specific killing rates, and effectively overcoming the resistance of *TP53*-mutant AML cells against CAR T-cells at the investigated E:T range (Fig. 6G,H).

In summary, genetic targeting of the negative regulator of the Wnt pathway Regnase-1 in CAR T-cells results in highly improved killing of *TP53*-deficient AML cells.

## Discussion

In this study, we addressed the emerging topic of genetic determinants of resistance to CAR T-cell therapies in hematological malignancies. We focused on *TP53*-mutant AML/MDS because current therapeutic approaches, i.e., intensive induction chemotherapy or HMAs+/− Ven have failed to improve patient outcomes (Pollyea et al, 2022), and thus, immunotherapeutic strategies are currently being tested in this clinically challenging patient cohort (Sallman et al, 2022). We took advantage of isogenic human AML cell lines harboring *TP53* null, missense, or wild-type alleles to exclude other potential genetic confounders. Such experimental approaches have successfully been employed to elucidate important aspects of the pathobiology of *TP53*-mutant AML/MDS (Boettcher et al, 2019; Sallman et al, 2020; Schimmer et al, 2022; Sellar et al, 2022; Thijssen et al, 2021). Our findings support a model, in which the intrinsic apoptotic defect in *TP53*-deficient AML/MDS cells results in a longer duration of the cellular interaction between CAR T-cells and *TP53*-deficient AML/MDS cells as shown by indirect as well as direct evidence of enhanced trogocytosis and live-cell imaging, respectively. Longer temporal

interaction between CAR T-cells and *TP53*-deficient AML/MDS leads to enhanced CAR T-cell exhaustion, reduced CAR T-cell proliferation, and eventually an overall decrease in AML cell killing and leukemia cell outgrowth (Fig. 7).

Although this is, to our knowledge, the first in-depth exploration of the impact of *TP53* deficiency in cancer cells on CAR T-cell-mediated killing, our data are supported by previous studies in physiologic cytotoxic T-cells (CTLs) and natural killer (NK)-cells. CTL- and NK-cell-mediated target cell apoptosis via perforin, granzyme B, or Fas-ligand was shown to be dependent on intact p53 in target cells (Ben Safta et al, 2015; Chollat-Namy et al, 2019; Thiery et al, 2005; Thiery et al, 2015).

Our results are also consistent with a recently published CRISPR-based genetic screen for antigen-independent CAR T-cell resistance mechanisms in acute B-lymphoblastic leukemia by Singh and colleagues, where the most-enriched sgRNAs belonged to the apoptosis pathway, with "p53 signaling" being in the top six most-enriched pathways (Singh et al, 2020).

The clinical evidence supporting a negative predictive role for *TP53* mutations in lymphoma in the context of anti-CD19 CAR T-cell therapies is, however, controversial. Shouval et al reported that *TP53* deficiency in large B-cell lymphoma (LBCL) led to inferior response and overall survival rates in patients treated with each of the clinically approved anti-CD19 CAR T-cell products (Shouval et al, 2022). By contrast, Jain et al could not confirm that *TP53* mutations are negative predictive biomarkers in LBCL patients treated with anti-CD19 CAR T-cells (Jain et al, 2022). This discrepancy might be explained by various confounding factors including co-mutations beyond *TP53*.

One limitation of our study may be the exclusive use of isogenic cell lines rather than primary cells. However, pre-clinical studies using well-defined isogenic cell lines model solely differing in the *TP53* mutational status appear ideally suited to rigorously addressing the question of the impact of individual gene mutations on therapy resistance. Of note, we observed highly consistent effects in two different isogenic AML cell line models. Our results, therefore, may help improve risk stratification and inform the design of future clinical trials of cellular therapies in *TP53*-mutant AML/MDS. Nevertheless, further studies are needed to confirm the clinical translatability of our findings—particularly in other cancer types.

Importantly, our data revealed both cancer cell-intrinsic as well as CAR T-cell-intrinsic pathways that could be exploited to overcome therapy resistance of *TP53*-mutant AML/MDS or to enhance therapeutic potency of CAR T-cells against *TP53*-mutant AML/MDS, respectively. Through transcriptional profiling of *TP53*-mutant AML cells under CAR T-cell attack we uncovered the mevalonate pathway to be most upregulated in comparison to

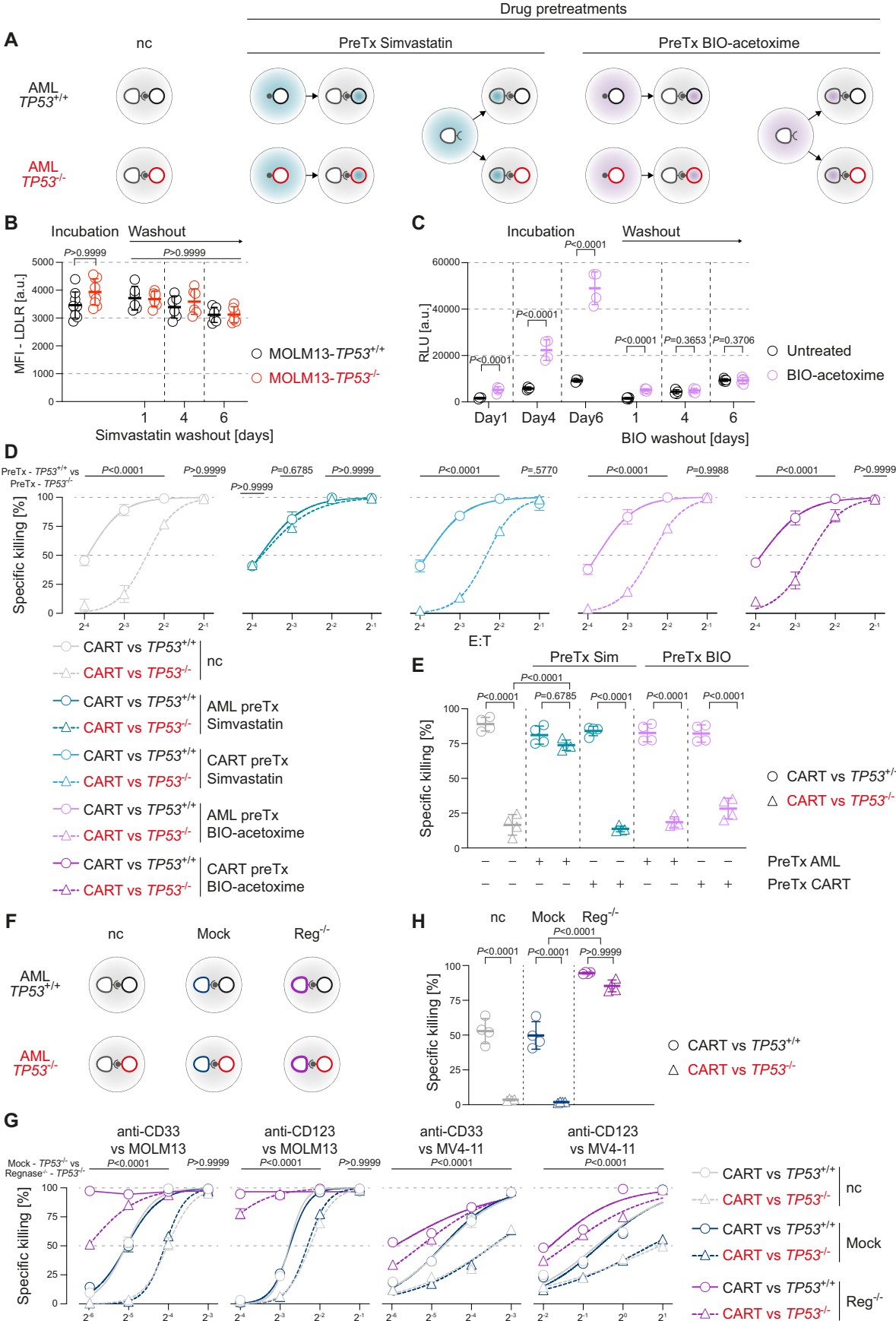

**Figure 6. Pretreatment with simvastatin but not with BIO-acetoxime rescues CAR T-cell killing of *TP53*-deficient AML cells.**

(A) Graphical representation outlining pretreatment co-incubation assays. (B) LDLR expression on the surface of MOLM13-*TP53* cells upon simvastatin 1 μM treatment for 24 h. (biological replicates, $n = 3$; two technical replicates per biological replicate; symbols indicate individual replicates; thickened line indicates mean and error bars indicate SD; two-way ANOVA). (C) Luminescence signal from T-cells transduced with a Wnt-responsive luciferase gene reporter system and incubated with or without BIO-acetoxime at 0.5 μM for 1, 4, or 6 days. (data shown from one healthy T-cell donor; biological replicates, $n = 2$; two technical replicates per biological replicate; symbols indicate individual replicates; thickened lines indicate mean and error bars indicate SD; unpaired Student's *t* test). (D) Results of pretreatment co-incubation assays with specific anti-CD33 CAR T-cell-mediated killing of MOLM13-*TP53*$^{+/+}$ or MOLM13-*TP53*$^{-/-}$ AML cells at various E:T ratios. (biological replicates, $n = 3$; two technical replicates per biological replicate; symbols represent means; error bars indicate SD; two-way ANOVA). (E) Summary data highlighting results from (D) at an E:T ratio of 1:8 (biological replicates, $n = 2$; two technical replicates per biological replicate; symbols represent individual replicates; thickened lines indicate mean and error bars indicate SD; two-way ANOVA). (F) Graphical representation outlining co-incubation assays with CRISPR/Cas9-edited CD33- and CD123-directed CAR T-cells against *TP53*$^{+/+}$ or *TP53*$^{-/-}$ AML cells. nc: unperturbed CAR T-cells; mock: CAR T-cells electroporated with nonsense gRNA and Cas9; RegKO: CAR T-cells gene-edited by CRISPR/Cas9 to induce Regnase-1 deficiency. (G) Results of co-incubation assays showing CRISPR/Cas9-edited CD33- and CD123-directed CAR T-cell-mediated killing (nc, mock and RegKO) against MOLM13-*TP53* and MV4-11-*TP53* AML cells. (biological replicates, $n = 2$; two technical replicates per biological replicate; symbols represent means; error bars indicate SD; two-way ANOVA). (H) Summary data highlighting results from (G) of gene-edited anti-CD33 CAR T-cell mediated killing against MOLM13-*TP53* at an E:T of 1:32 (biological replicates, $n = 2$; two technical replicates per biological replicate; symbols represent individual replicates; thickened lines indicate mean and error bars indicate SD; two-way ANOVA). Source data are available online for this figure.

*TP53* wild-type AML cells. Accordingly, pharmacological inhibition of HMG-CoA reductase— the rate-limiting enzyme of the mevalonate pathway—using simvastatin could fully rescue CAR T-cell-mediated killing of *TP53*-mutant AML cells. These results corroborate and extend the existent literature on the impact of mevalonate pathway in cancer (Mullen et al, 2016). p53 is a known negative regulator of the mevalonate pathway with increased cholesterol levels shown to promote tumorigenesis in a pre-clinical cancer models of breast carcinoma and hepatocellular carcinoma (HCC) (Freed-Pastor et al, 2012; Moon et al, 2019). On the molecular level, decreasing the cholesterol content of a target cell was shown to increase the stiffness of its cell membrane, which in turn increased the perforin-dependent cell death and killing by cytotoxic T-cells (Basu et al, 2016; Lei et al, 2021). Moreover, several cohort studies and meta-analyses, comprising more than 600,000 cancer patients, have consistently found a small but statistically significant prolongation of overall survival and/or disease-specific survival associated with statin treatment (Bansal et al, 2012; Cardwell et al, 2015; Gray et al, 2016; Huang et al, 2017; Manthravadi et al, 2016; Nayan et al, 2017; Nielsen et al, 2012; Peng et al, 2015). One caveat is, however, that the typical plasma concentrations achieved with oral simvastatin therapy are about 2–3-log-fold lower than the one we used in the here reported in vitro killing assays (Bjorkhem-Bergman et al, 2011; Keskitalo et al, 2009). Therefore, prohibitive dose-limiting toxicities will prevent immediate clinical translation of our pre-clinical findings. Whether more specific inhibition of the mevalonate pathway in *TP53*-deficient AML/MDS in the context of CAR T-cell therapies might provide a therapeutic window remains to be determined and will be the focus of future studies.

With regards to CAR T-cell-intrinsic resistance pathways, transcriptional profiling of CAR T-cells attacking *TP53*-deficient AML cells revealed a significant downregulation of the transcription factors *TCF7* and *EOMES*, both of which are signaling downstream of the canonical Wnt pathway, and are essential in T-cell development, differentiation, and survival (Staal et al, 2004; Zhou et al, 2010). These data strongly suggest that insufficient activation of the Wnt pathway in CAR T-cells may contribute to their reduced activity against *TP53*-deficient AML/MDS. Accordingly, our data demonstrate that sustained activation of the Wnt pathway by pharmacological intervention or by overexpression of *TCF7* via knocking out its negative regulator Regnase-1, effectively overcomes *TP53* deficiency-associated CAR T-cell resistance, and further enhances CAR T-cell efficacy.

Last, our study underscores the importance of achieving favorable E:T ratios in a therapeutic setting, as has been observed in clinical trials (Ali et al, 2016; Enblad et al, 2018; Milone and Bhoj, 2018; Raje et al, 2019; Turtle et al, 2016a; Turtle et al, 2016b). By increasing the in vitro E:T ratio we could eventually eradicate *TP53*-deficient AML cells and observed similar exhaustion and differentiation profiles as in CAR T-cells engaging isogenic, *TP53*-proficient AML cells at lower E:T ratios. Our data suggest that treatment of patients with *TP53*-deficient hematological neoplasms will require a relatively higher dose of infused CAR T-cells at a given tumor load (e.g., minimal residual disease) to achieve complete remissions—an approach, which has been pursued with other malignancies stratified as a high-risk disease (Brudno et al, 2018). However, this may come at the cost of increased toxicity (Dasyam et al, 2020; Milone and Bhoj, 2018). Therefore, the combination of CAR T-cell therapies with pharmacological co-interventions or the use of genetically enhanced CAR T-cell products—as exemplified in this study—may be a preferable strategy towards more efficacious and tolerable cellular therapies for patients with *TP53*-mutant myeloid neoplasms.

## Methods

### AML cell lines

Isogenic MOLM13 cells harboring wild-type *TP53* (MOLM13-*TP53*$^{+/+}$), knockout *TP53* (MOLM13-*TP53*$^{-/-}$) or missense mutant (MOLM13-*TP53*$^{missense}$) alleles (Boettcher et al, 2019) were cultured in R10 (Roswell Park Memorial Institute media (RPMI) supplemented with GlutaMAX™ (Gibco™, #61870010), 100 U/ml of penicillin and 100 μg/ml of streptomycin (penicillin–streptomycin 5000 U/ml, Cat. No. 15070063), and 10% FCS (Gibco™, #16000044)). MV4-11 cells (CRL-9591™, ATCC®, Manassas, VA, USA) were cultured in Iscove's Modified Dulbecco's Medium (IMDM), supplemented with 10% FBS and 100 U/ml of penicillin and 100 μg/ml of streptomycin. All cells were kept in a humidified atmosphere of 95% air and 5% $CO_2$ at 37 °C. Cells were passaged every 3–5 days. Aliquots of early passages (P3-P4) were cryopreserved and thawed every 2–3 months for experiments to ensure low passages.

For competition experiments, established cell lines with stable fluorophore expression, MOLM13-*TP53*$^{+/+}$RFP$^+$ and

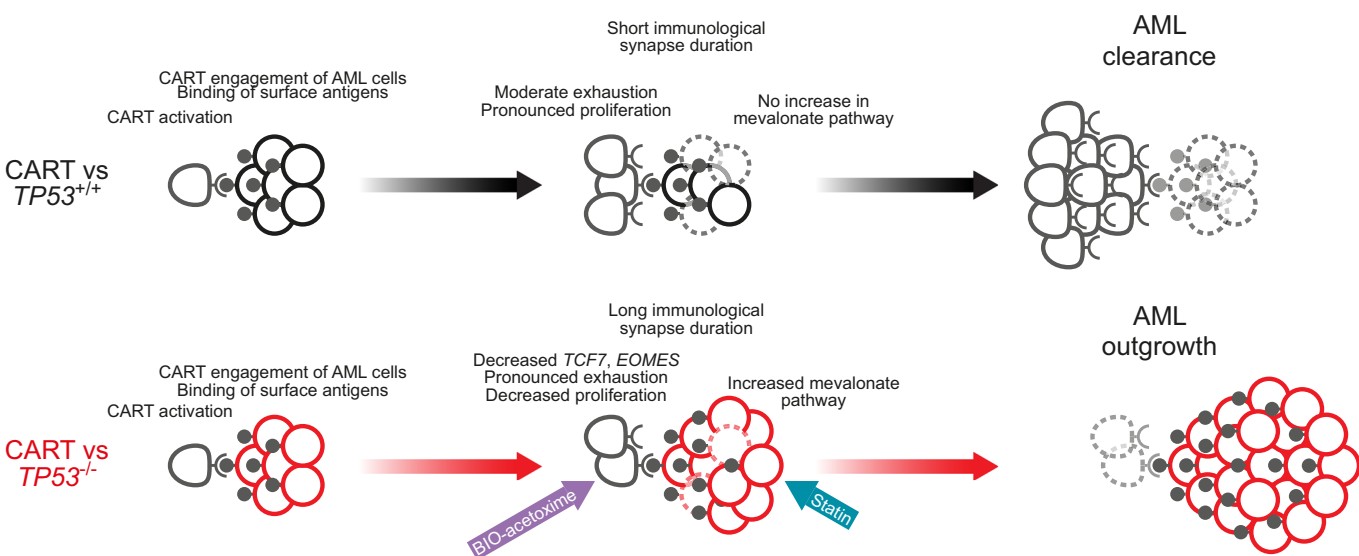

**Figure 7. *TP53* deficiency in AML leads to resistance against CAR T-cell-mediated killing that can be overcome by targeting the mevalonate and Wnt pathways.**

Graphical representation of our proposed model. The intrinsic apoptotic defect in *TP53*-deficient AML cells leads to a longer duration of cellular interaction between CAR T-cells and *TP53*-deficient AML cells. Longer temporal interaction between CAR T-cells and *TP53*-deficient AML eventually leads to reduced CAR T-cell proliferation and enhanced CAR T-cell exhaustion resulting in the outgrowth of *TP53*-deficient AML cells. Inhibition of cholesterol biosynthesis in *TP53*-deficient AML cells or activation of the Wnt pathway in CAR T-cells can overcome resistance of *TP53*-deficient AML cells to CAR T-cell-mediated killing.

MOLM13-*TP53*$^{-/-}$GFP$^+$, were used as previously described (Boettcher et al, 2019).

All cells were regularly tested for mycoplasma contamination (Mycostrip™—Mycoplasma Detection Kit, InvivoGen, Cat. No. rep-mys-10).

## Production of lentiviral vectors

Replication-defective, third-generation lentiviral vectors were produced as previously described (Myburgh et al, 2020). In brief, HEK293T were transfected with the transfer plasmid containing the DNA sequence of interest as well as with psPAX2 packaging and pCAG-VSVG envelope plasmids (both kindly provided by Dr. Patrick Salmon, University of Geneva, Switzerland). JetPRIME® transfection reagent (Polyplus transfection, Illkirch, France) was used to enhance transfection efficacy. Viral particles were harvested 2 days later and concentrated with Peg-it™ (System Biosciences, Palo Alto, CA, USA). Concentrated viral particles were stored at −80 °C until further use.

## Generation of MOLM13-*TP53* cells expressing GFP and luciferase

Lentiviral transduction of MOLM13-*TP53* with Luciferase pCLX-UBI-GFP-Luc (kindly provided by Dr. Patrick Salmon, University of Geneva, Switzerland) was performed as described before (Myburgh et al, 2020) in the presence of hexadimethrine bromide (8 µg/ml, Sigma-Aldrich, St. Louis, MO, USA) by spin infection. Subsequent cell sorting of GFP+ populations on a FACS Aria III 4L (Becton Dickinson, Franklin Lakes, NJ, USA) of oligoclonal populations yielded pure MOLM13-*TP53*$^{+/+}$Luc$^+$ and MOLM13-*TP53*$^{-/-}$Luc$^+$ populations. The resulting clones were kept in culture

and monitored for growth kinetics. Stable clones were selected and cryopreserved for future experiments.

## Generation of gene knockout AML cell lines using CRISPR/Cas9

MV4-11-*TP53*$^{-/-}$ were generated as previously described (Schimmer et al, 2022). In summary, oligonucleotides for the cloning of sgRNAs targeting human *TP53* were synthesized (Microsynth AG, Switzerland; Human *TP53* KO top oligo, CACCGTCGGATAA-GATGCTGAGGAG; Human *TP53* KO bottom oligo, AAACCTCCTCAGCATCTTATCCGAC), annealed and cloned into pL-CRISPR.SFFV.GFP (pL-CRISPR.SFFV.GFP was a gift from Benjamin L. Ebert (Addgene plasmid #57827; http://n2t.net/addgene:57827; RRID:Addgene 57827)). Plasmids were then propagated in Sblt3 bacterial strains and lentiviruses were subsequently produced as previously described (Myburgh et al, 2020). Empty plasmid vectors expressing Cas9 and tag-RFP were propagated and lentiviruses produced to be used as controls (pL-CRISPR.SFFV.tRFP was a gift from Benjamin L. Ebert (Addgene plasmid #57826; http://n2t.net/addgene:57826; RRID: Addgene_57826)). The human AML cell line MV4-11 was transduced with lentiviral supernatants of either the vector containing the *TP53* knockout sgRNA sequence with GFP or with the empty vector carrying Cas9 only and tag-RFP. Fluorochrome-expressing cells were sorted using a FACS Aria 4L (Becton Dickinson, Franklin Lakes, NJ, USA) to obtain a polyclonal CRISPR/Cas9-edited cell population. Functional loss of *TP53* protein expression was confirmed by immunoblotting for p53 and p21 following culture of cells for 6 h in the presence of the MDM2 inhibitor Nutlin-3a (10 µM; Selleckchem, Cat. No. S8059; see Appendix Fig. 6A) and Daunorubicin (100 µM; Selleckchem,

Cat. No. S3035; see Appendix Fig. 6A) as well as via apoptosis assay using Annexin V Allophycocyanin (APC) staining (Biolegend, Cat. No. #640920) in combination with DAPI (4',6-diamidino-2-phenylindole dilactate; Biolegend, Cat. No. # 422801; see Appendix Fig. 6B).

## Immunoblots

MV4-11-*TP53* cells were seeded at a concentration of $1.5 \times 10^6$ cells/ml in six-well plates and incubated with DMSO, Nutlin-3a (10 µM; Selleckchem, Cat. No. S8059) or Daunorubicin (100 µM; Selleckchem, Cat. No. S3035) for 6 h. Whole-cell extracts were prepared using the PierceTM IP Lysis Buffer (ThermoFischer, Cat. No. 87787) freshly supplemented with Protease Inhibitor (cOmpleteTM Protease Inhibitor Cocktail, Roche). Protein concentration was quantified using PierceTM BCA Protein Assay Kit (ThermoFischer, Cat. No. 23225/23227) and absorbance was measured at 562 nm. Equal protein amounts were loaded and run on 4–12% gradient gels, transferred to PVDF membranes (Invitrogen, Cat. No. STM2006) and subjected to immunoblotting using primary, followed by corresponding HRP-conjugated secondary antibodies. Primary and secondary antibodies used in this study: vinculin (rabbit monoclonal antibody ERP8185, Abcam, Cat. No. 129002; dilution 1:5000), p53 (mouse monoclonal antibody DO-1, Cell Signaling Technology, Cat. No. 18032; dilution 1:1000), p21 (rabbit monoclonal antibody 12D1, Cell Signaling Technology, Cat. No. 2947; dilution 1:1000), goat anti-mouse-HRP (Genesee Scientific, Cat. No. 20-304, dilution 1:5000), goat anti-rabbit-HRP (Genesee Scientific, Cat. No. 20-303, dilution 1:5000). Blots were visualized using Immobilon Forte HRP substrate (Merck, Cat. No. WBLUF0100) and developed using a Stella 3200 (Elysia-Raytest, Angleur, Belgium). Immunoblots were repeated in three biological replicates.

## Cloning of lentiviral CAR expression vectors

The published sequences of the scFv domains targeting CD33 (clone SGN-33, Lintuzumab; Feldman et al, 2005), CD123 (clone H9; Hutmacher et al, 2019), CD123 (clone CSL362; Lee et al, 2015b), and CD117 (clone D79; Myburgh et al, 2020) were cloned into a lentiviral expression vector (pCDH-EF1α-MCS-T2A-GFP, System Biosciences, Palo Alto, CA, USA) to yield transgenes coding for four different second-generation CARs and the RQR8 peptide (Philip et al, 2014) (Fig. EV1A). The resulting plasmids were packaged into lentiviral vectors to produce viral particles for transduction as described above.

## CAR T-cells

Healthy donor (HD) buffy coats were obtained from the blood donation service Zurich (Blutspende Zürich, Switzerland), T-cells were then negatively selected with EasySepTM beads (Stemcell Technologies, Vancouver, Canada, #17951). T-cells were generally kept at a density of $10^6$/ml in T-cell medium (Advanced RPMI 1640 (Gibco™, #12633012) supplemented with penicillin (penicillin (100 U/ml)/streptomycin (100 µg/ml), GlutaMax™ (1:100, Gibco, # 35050061) and 10% FBS (Gibco™, #16000044) unless stated otherwise. T-cell and CAR T-cell stimulation, expansion, and purification by MACS was performed as previously described (Myburgh et al, 2020). Briefly, negatively selected healthy donor

T-cells were thawed, seeded at $2 \times 10^6$/ml in T-cell medium for 24 h and stimulated with CD3/CD28 Dynabeads (40,000/µl, Gibco™, #11161D) at a 1:1 ratio for 3 days in the presence of IL-2 (100 U/ml, Gibco™, #CTP0021). T-cell transduction with concentrated lentiviral particles carrying the above-described CAR transgenes was performed 1 day after the start of the stimulation at an MOI of 3–4 in the presence of 8 µg/ml hexadimethrine bromide (Sigma-Aldrich, St. Louis, MO, USA, #H9268) for 16 h. For untransduced controls, the protocol was performed in an identical manner without adding lentiviral vectors. A typical resulting batch of a MACS-purified CAR T-cell population is shown in Appendix Fig. S1B. Expanded untransduced T-cells and CAR T-cells of different healthy donors were cryopreserved and stored at liquid nitrogen for further use.

## In vitro co-culture assays

For in vitro killing assays, purified CAR T-cells were co-cultured with isogenic MOLM13-*TP53* or MV4-11-*TP53* cells in round-bottom 96-well plates at effector-to-target (E:T) ratios ranging from $2^1$:1 to $2^{-8}$:1 at a total cell number of 60,000 cells per well in R10 cell culture medium at 37 °C and 5% $CO_2$. Cells were split every 2 days, supplemented with new R10 medium and kept for up to 10 days, aliquots of the cell culture were used for intermittent flow cytometric analyses as described below. If not stated otherwise, experiments were generally performed with three different healthy donors in technical duplicates and at least two biological replicates. Co-culture assays for mRNA-seq were set up in 96-well plates using three full plates for every condition and using the same healthy donor untransduced T-cells and CAR T-cells for the three technical replicates started at different time points. For drug-treated in vitro co-culture assays, the addition of simvastatin, BIO-acetoxime and/or DL-Mevalonolactone was achieved using an HP® D300e Digital Dispenser (HP Inc.) at designated concentrations. Cells were incubated for a total of 6 days, at which flow cytometric analysis was performed.

For pretreatment co-culture assays, targets and effectors were seeded in six-well and 12-well plates, respectively, at a concentration of $1 \times 10^6$ cells/ml and exposed to simvastatin and/or BIO-acetoxime at indicated concentrations for 24 h. In addition, IL-2 (100 U/ml, Gibco™, #CTP0021) was added to effectors during pretreatment period. Cells were then collected, washed with PBS and then seeded for co-culture assays as described above.

## Flow cytometry

For surface staining, FACS antibodies listed in Appendix Table S1 were used at dilutions ranging from 1:20-1:800, generally at 20 min at 4 °C. Hoechst 33342 (20 mM, Thermo Scientific, Waltham, MA, USA, #62249), Fixable Viability Dye (FVD) eFluor™ 780 (eBioscience™, San Diego, CA, USA) or DAPI (4',6-Diamidino-2-Phenylindole Dilactate; Biolegend, Cat. No. # 422801) were used for live/dead discrimination at 1:5000, 1:2000, or 1:2500 dilutions, respectively. Data were acquired on an LSRFortessa™ 4L equipped with a high-throughput sampler (HTS) (Becton Dickinson, Franklin Lakes, NJ, USA). Data were analyzed and quantified using the FlowJo® v9x software (Becton Dickinson, Franklin Lakes, NJ, USA). Cell sorting for the establishment of pure cell lines or for mRNA sequencing was performed on an Aria III 4L (Becton Dickinson Franklin Lakes, NJ, USA).

## Live-cell imaging

Time-lapse experiments were performed at 37 °C, 5% $CO_2$ in R10 growth medium supplemented with propidium iodide (2 µM, eBioscience™, # BMS500PI) on µ-Slides with four wells (Ibidi®, #80426). Self-adherent silicone polymer microgrid arrays with 50 µm × 50 µm dimensions (Microsurfaces, Flemington, Australia, #MGA-050-02) were glued to the bottom of each well to allow for separate observation of individual killing events. CD33-directed CAR T-cells were stained with CellVue® Claret Far Red fluorescent membrane dye (Sigma-Aldrich®, #C0744) in Diluent C (Sigma-Aldrich®, #G8278) according to the manufacturer's instructions prior to co-incubation. MOLM13-TP53$^{+/+}$ or MOLM13-TP53$^{-/-}$ were used as target cells at an E:T ratio of 1:1 with 40,000 cells per well. Time-lapses were conducted on a Nikon-Ti Eclipse equipped with a linear-encoded motorized stage, Orca Flash 4.0 V2 (Hamamatsu), a Spectra X fluorescent light source (Lumencor), a 10× CFI Plan Apochromat λ objective (NA 0.45) and an incubation chamber. Appropriate filter sets (all AHF) were chosen to detect fluorescence: GFP (470/40; 495LP; 525/50), CellVue® Claret Far Red (620/60; 660LP; 700/75) and propidium iodide (550/32; 585LP; 605/15). Bright-field images were acquired using white light emitted by the Spectra X and a custom-made motorized mirror controlled by Arduino UNO Rev3 (Arduino). Typically, 16 positions per microgrid were acquired at 10 min time intervals for up to 96 h. Single-cell tracking was performed using custom-written software (Hilsenbeck et al, 2016; Hoppe et al, 2016; Loeffler et al, 2019), with CAR T-cell to AML cell attachment until AML cell or CAR T-cell death as judged by propidium iodide influx or detachment of CAR T-cell from the AML cell scored for each individual attachment event. Three different healthy donor CAR T-cells were used as three biological replicates. Events per group: CAR T-cells vs. MOLM13-TP53$^{+/+}$ = 102, CAR T-cells vs. MOLM13-TP53$^{-/-}$ = 88.

## In vivo xenograft model

All experiments and procedures involving experimental animals were performed according to protocols approved by the Cantonal Veterinary Office Zurich (Kantonale Veterinäramt Zürich) under the license 194/2018 as well as to the ARRIVE guidelines. Animals were maintained under specific pathogen-free (SPF) conditions at the Laboratory Animal Services Center (LASC, Schlieren, Zurich, Switzerland) and allowed to acclimatize for 1 week before use in experiments. Mice were kept in individually ventilated cages (IVC type 2 long; IVC T2L) housing up to five mice per cage with standard housing enrichment (bedding, red mouse house, tissues, and crinklets). The animal facility is equipped with an automated watering system with a connected drinking valve providing fresh water ad libitum. A schematic outline of the murine xenograft leukemia model is presented in Fig. 4A. Female 8–12 weeks old NOD-Prkdc$^{scid}$-IL2rg$^{Tm1}$/Rj mice (Janvier Labs®, Le Genest-Saint-Isle, France) were sublethally irradiated with 100 cGy using an RS-2000 irradiator (Rad Source, Buford, GA, USA). After 6 h mice were injected with 10$^5$ MOLM13-TP53$^{+/+}$Luc$^+$ or MOLM13-TP53$^{-/-}$Luc$^+$ cells via tail vein injection. A sufficient dose of CD33-directed T-cells to control leukemic disease for at least 3 weeks was established by pilot experiments for three different healthy donor CAR T-cells. Accordingly, 2–8 × 10$^6$ anti-CD33 CAR T-cells of the respective healthy donors were injected 7 days after MOLM13-TP53 AML cell engraftment via tail vein injection. Bioluminescence measurements were conducted on the indicated days by intraperitoneal injection of D-Luciferin (250 mg/kg bodyweight, Xenolight, PerkinElmer®, Waltham, MA, USA, #122799). Time points for the summary analyzes shown in Fig. 4C were chosen at day 7 and the latest day with data from all mice (before occurrence of termination criteria in any mouse, see below), corresponding to day 15 or day 18 after MOLM13-TP53 AML injection. Mice were monitored for signs of disease and sacrificed by $CO_2$ euthanasia if any of the following termination criteria were observed: paralysis of any leg, moribund appearance (unresponsive to stimuli, labored respiration, limited ambulation), body weight loss >15–20% of starting weight or dehydration unresponsive to Glucose 5% injections. Part of the mice were sacrificed at the day 35 without showing signs of disease for terminal analysis. Pooled survival data of three biological replicates with three different healthy donors are shown in Fig. 4D. Animal counts per group: T-cells vs. MOLM13-TP53$^{+/+}$Luc$^+$=12, T-cells vs. MOLM13-TP53$^{-/-}$Luc$^+$=11, CAR T-cells vs. MOLM13-TP53$^{+/+}$Luc$^+$=15, CAR T-cells vs. MOLM13-TP53$^{-/-}$Luc$^+$=15.

## mRNA sequencing

Samples for gene expression profiling by mRNA-seq were set up as described above in technical triplicates in three 96-well plates per condition with CD33-directed CAR T-cells of the same healthy donor and MOLM13-TP53$^{+/+}$ or MOLM13-TP53$^{-/-}$ at an E:T of 2$^{-4}$:1 for 6 days. After co-incubation, CAR T-cells and MOLM13-TP53 AML cells were sorted using a FACS Aria III 4L (Becton Dickinson, Franklin Lakes, NJ, USA). Details of the gating strategy and sorting results are provided in Fig. EV3. For mRNA extraction and isolation were performed using the PicoPure™ RNA isolation kit (ThermoFisher, #KIT0204) according to the manufacturer's instructions. Cells were directly sorted into extraction buffer and stored at −80 °C until further processing. High-throughput next-generation sequencing was performed at the Functional Genomics Center Zurich on a NovaSeq 6000 Sequencing System (Illumina®, San Diego, CA, USA). Quality control, mapping and differential gene expression analysis was performed using the SUSHI bioinformatics pipeline. Gene set enrichment analysis (GSEA) was performed using the GSEA software jointly developed by UC San Diego and the Broad Institute (Subramanian et al, 2005) and the Enrichr tool (Chen et al, 2013; Kuleshov et al, 2016; Xie et al, 2021). The curated gene set WP_CHOLESTEROL_BIOSYNTHESIS_-PATHWAY was used for the analysis of expression of genes involved in cholesterol biosynthesis.

## Chemical compounds

The GSK-3 inhibitor BIO-acetoxime (Selleckchem, Cat. No. #S7915) and simvastatin (Selleckchem, Cat. No. #S1792) were ordered in powder formulation. DL-Mevalonolactone (Medchemexpress, Cat. No. HY-107855) was ordered in oil formulation. BIO-acetoxime and DL-Mevalonolactone were dissolved or diluted in Dimethyl Sulfoxide (DMSO) respectively, aliquoted and stored at −80 °C. Simvastatin was activated by alkaline hydrolysis according to the manufacturer's protocol (Sadeghi et al, 2000). Briefly, the lactone pro-drug was dissolved in 95% ethanol at a pH of 1.0 at 50 °C for 2 h before being equilibrated to a pH of 7.2, upon which the drug was aliquoted and stored for further use at −20 °C. Drugs were always freshly thawed and diluted appropriately either in cell culture medium (for in vitro studies), DMSO (for digital dispensing of BIO-acetoxime and DL-Mevalonolactone) or PBS-Tween 0.3%

(for digital dispensing of simvastatin) to obtain desired concentrations.

## Production of CRISPR/Cas9-edited CAR T-cells

Single-guide RNA (sgRNA) sequence targeting Regnase-1 was previously published (Mai et al, 2023; see Appendix Table S2). Recombinant Cas9 protein (Cat. No. #1081058), synthetic locus-specific CRISPR RNA (crRNA), and transactivating crRNA (tracrRNA, Cat. No. #1072532) were all purchased from Integrated DNA Technologies (IDT™). Equimolar amounts (120 pmol) of crRNAs and tracrRNAs were mixed in a Cas9 buffer (HEPES 20 mM, 150 mM KCl, 1 mM $MgCl_2$, 10% glycerol, 1 mM TCEP) to obtain a total volume of 25 µl. To generate crRNA:tracrRNA duplexes, the mixtures were heated to 98 °C for 5 min and allowed to cool down to ambient temperature. Recombinant Cas9-3NLS (100 pmol) was diluted in Cas9 working buffer to obtain a final volume of 25 µl. Both crRNA:tracrRNA duplexes and diluted Cas9-3NLS were slowly mixed and incubated for 20 min at room temperature to allow the formation of ribonucleoprotein (RNP) complexes. CRISPR/Cas9-editing of T-cells was achieved using the human T-cell Nucleofector Kit (Lonza, Cat. No. #VPA-1002) and according to the manufacturer's protocol. Briefly, upon CD3/CD28-bead removal on day 3 of CAR T-cell production (as described above), $3–5 \times 10^6$ transduced T-cells were harvested, spun down, and resuspended in 50 µl of nucleofection solution. RNP and cell solutions were combined in a 100 µl nucleocuvette. Electroporation was carried out using a 2B-Nucleofector (Lonza) using cell line-specific settings according to the manufacturer's recommendations. Electroporated T-cells were resuspended at a density of $1 \times 10^6$ cells/ml in T-cell medium (Advanced RPMI 1640 (Gibco™, #12633012) supplemented with penicillin (100 U/ml)/streptomycin (100 µg/ml), GlutaMax™ (1:100, Gibco, # 35050061) and 10% FBS (Gibco™, Cat. No. #16000044) in addition to IL-2 (100 U/ml, Gibco™, Cat. No. #CTP0021).

## CRISPR sequencing

Genome editing was assessed by ultra-deep amplicon next-generation sequencing performed at the Massachusetts General Hospital Center for Computational and Integrative Biology DNA core facility (MGH CCIB DNA core, Cambridge, MA). For this, DNA was extracted from cells using the QIAamp DNA Mini Blood Kit (QIAgen, Cat. No. 51304) and subjected to PCR amplification of genomic knockout regions (Phusion® High-Fidelity PCR Master Mix with GC Buffer, New England BioLabs® Inc, Cat. No. M0532S). PCR-amplified DNA was then purified (QIAquick PCR Purification Kit, QIAgen, Cat. No. 28104) and sent for sequencing. Purification was visualized by agarose gel electrophoresis (data not shown).

## Quantitative reverse-transcription PCR

Samples for gene expression profiling by RT-qPCR were set up in analogy to mRNA-seq experiments: for each condition a full 96-well plate with CD33- or CD123-directed CAR T-cells of the same healthy donor and MOLM13-$TP53^{+/+}$/MOLM13-$TP53^{-/-}$ or MV4-11-$TP53^{+/+}$/MV4-11-$TP53^{-/-}$ were incubated for 6 days. E:T ratios for respective co-cultures can be found in Appendix Table S3. After co-incubation, CAR T-cells and MOLM13/MV4-11-$TP53$ AML cells were sorted using a FACS Aria III 4L (Becton Dickinson, Franklin Lakes, NJ, USA). Cells were sorted into FACS Buffer, spun down and pellets were stored at −80 °C until further processing. Total RNA isolation was carried out using the RNeasy kit (QIAGEN, Cat. No. #74104) according to the manufacturer's instructions. Reverse transcription was done using SuperScript™ IV VILO™ Master Mix with ezDNase™ Enzyme (ThermoFisher, Cat. No. #11766500). qPCR was performed using TaqMan® Gene Expression Master Mix (ThermoFisher, Cat. No. #4369016) and TaqMan® Gene Expression Assays (HMGCR, Hs00168352_m1; HMGCS1, Hs00940429_m1; TCF7, Hs01556515_m1; EOMES, Hs00172872_m1; ACTB, Hs01060665_g1) on a 7900HT Fast-Real-Time PCR System (Applied Biosystems).

## LDLR expression experiments

MOLM13-$TP53$ and MV4-11-$TP53$ cells were seeded in 96-well flat-bottom cell culture plates at a density of $1 \times 10^5$ cells/well. Cells were treated with (titrated) simvastatin and/or DL-mevalonolactone at designated concentrations and staining with mouse anti-human LDLR PE (BD Pharmingen™, Cat. No. 565653) was performed to determine mean fluorescence intensity using BD LSRFortessa™ Flow Cytometer with the high-throughput sampler (HTS; Becton Dickinson, Franklin Lakes, NJ, USA) at different time points. Drug application was performed using an HP® D300e Digital Dispenser (HP Inc.).

## Luciferase gene reporter assays

For gene reporter assays, plasmids containing SRE- and Wnt-responsive promoters coupled to luciferase were purchased from Addgene (Addgene plasmid #90371: pLminP_Luc2P_RE29 for SRE reporter assays and Addgene plasmid #24307: 7TFC for Wnt reporter assays). Plasmids were propagated, and lentiviral vectors produced as described above. For SRE reporter experiments, MOLM13-$TP53$ and MV4-11-$TP53$ were transduced with lentiviral supernatants of pLminP_Luc2P_RE29 vectors, sorted for fluorochrome-expressing cells using a FACS Aria 4L (Becton Dickinson, Franklin Lakes, NJ, USA) and cryopreserved in aliquots for further experiments (MOLM13-$TP53$-SRE and MV4-11-$TP53$-SRE). Similarly, for Wnt reporter experiments, T-cells isolated from two different healthy donors were thawed, stimulated overnight with CD3/CD28 beads as described above and then transduced with lentiviral supernatants of 7TFC vectors. After bead removal on day 3, T-cells were then propagated in the presence of IL-2 (100 U/ml, Gibco™, Cat. No. #CTP0021), sorted for fluorochrome-positive cells and expanded until day 10 after thawing. 7TFC expressing T-cells were then cryopreserved in batches for further experiments (7TFC-T-cells).

MOLM13-$TP53$-SRE, MV4-11-$TP53$-SRE and 7TFC-T-cells were seeded in opaque-walled 96-well flat-bottom culture plates and incubated with increasing concentrations of simvastatin (in the presence or absence of Mevalonate 1 mM) or BIO-acetoxime, respectively. MOLM13-$TP53$-SRE cells were seeded at a density of $1 \times 10^5$ cells/well, MV4-11-$TP53$-SRE were plated at $0.5 \times 10^5$ cells/well and freshly thawed 7TFC-T-cells were seeded at a density of $1 \times 10^5$ cells/well in the presence of IL-2. Limiting dilutions of simvastatin and BIO-acetoxime were prepared using an HP® D300e Digital Dispenser (HP Inc.). At designated time points, cells were

## The paper explained

### Problem

*TP53*-mutant acute myeloid leukemia/myelodysplastic neoplasms (AML/MDS) are distinct clinicogenomic entities characterized by chemotherapy resistance, high relapse rates, and poor survival. Chimeric antigen receptor (CAR) T-cell therapy is a successful novel cell-based therapy in hemato-oncology, and it might also be a promising therapeutic option for *TP53*-mutant AML/MDS. However, the AML-intrinsic determinants of the efficacy of CAR T-cell-based therapies are largely unknown.

### Results

Our study shows that *TP53* mutations in AML cells lead to increased resistance to CAR T-cells in vitro. CAR T-cells co-incubated with target *TP53*-mutant AML blasts exhibited decreased proliferation and increased exhaustion compared to wild-type *TP53* AML blasts. Live-cell imaging revealed longer time-to-killing of *TP53*-mutant than wild-type *TP53* AML cells upon attack by CAR T-cells, which ultimately led to an inability of CAR T-cells to control *TP53*-mutant cells. Furthermore, immunodeficient mice xenografted with *TP53*-mutant AML and treated with CAR T-cells exhibit shorter survival compared to mice engrafted with wild-type *TP53* AML. Transcriptional profiling of AML cells with either wild-type or mutant *TP53* under CAR T-cell attack revealed upregulation of the mevalonate synthesis pathway in *TP53*-mutant AML cells. Simultaneously, CAR T-cells engaging *TP53*-mutant AML demonstrated a downregulated Wnt pathway. Rational pharmacological targeting of either of these pathways rescued *TP53*-mutant AML cell sensitivity to CAR T-cell-mediated killing. Similarly, CRISPR/Cas9-engineering of CAR T-cells to upregulate Wnt pathway signaling rescued *TP53*-mutant AML cell sensitivity and led to improved killing efficacy.

### Impact

We demonstrate that *TP53* deficiency in AML cells confers resistance to CAR T-cell therapy by inducing CAR T-cell dysfunction. We propose a model in which difficult-to-kill *TP53*-mutant AML cells promote CAR T-cell exhaustion, eventually leading to uncontrolled AML cell outgrowth. We further identify inhibition of the mevalonate pathway as a potential therapeutic vulnerability of *TP53*-mutant AML cells, and stimulation of the Wnt pathway as a promising avenue to enhance the efficacy of CAR T-cell therapy. Pharmacological co-interventions or genetic engineering of CAR T-cell products may thus be a promising strategy for more efficacious and tolerable cellular therapies for patients with *TP53*-mutant AML/MDS.

spun down, culture medium was removed and cell pellets were resuspended in 100 µl lysis buffer (25 mM Tris-phosphate at pH 7.8, 2 mM DTT, 2 mM 1,2-diaminocyclohexane-N,N,N′,N′-tetra-acetic acid, 10% glycerol, 1% Triton® X-100). Luciferase activity was assessed by adding 100 µl of 2X D-Luciferin (Xenolight D-Luciferin K⁺ Salt Bioluminescent Substrate, Cat. No. 122799, PerkinElmer, Inc, Waltham, MA, USA) to lysates and luminescent signal was acquired using a Biotek Synergy LX (SLXFTS) luminometer after a 5-min incubation period at room temperature in the dark. Experiments were performed in duplicates and repeated in at least two biological replicates.

## Statistical analysis

Statistical significance testing was determined by the indicated tests using GraphPad Prism 8 (GraphPad Software Inc., San Diego, CA,

USA). For in vivo experiments, no blinding or randomization occurred. All animals were included for the final analysis (no exclusion).

## Data availability

RNA-seq data: Gene Expression Omnibus GSE246335. Live-cell imaging data: BioStudies repository S-BIAD926.

## Peer review information

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

## Acknowledgements

This work was supported by research grants of the ETH Zurich made available through the support of Dr. Walter & Edit Fischli to TS and MGM, a University Research Priority Project Translational Cancer Research grant to RM, and the Clinical Research Priority Program "ImmunoCure" of the University of Zurich to MGM, the KRAK—Physician Scientist Fellowship and the Jacques and Gloria Gossweiler Foundation to RRS, the KRAK—Physician Scientist Fellowship to JM, as well as research grants by the Promedica Foundation Chur, the Swiss Cancer League (KFS-4885-08-2019), the Fondation Peter Anton & Anna Katharina Miescher pour la Recherche en Hématologie, and the Swiss Society of Hematology to SB. The authors thank the Functional Genomics Center Zurich and the Laboratory Animal Services Center (LASC) for their support. Furthermore, JM would like to thank Dr. Philipp Altrock, Saumil Shah, and Dr. Eva Avilla Royo for critical discussions and their support in setting up experiments.

## Author contributions

**Jan Mueller**: Conceptualization; Data curation; Formal analysis; Validation; Investigation; Visualization; Methodology; Writing—original draft; Writing—review and editing. **Roman R Schimmer**: Conceptualization; Resources; Data curation; Formal analysis; Validation; Investigation; Visualization; Methodology; Writing—original draft; Writing—review and editing. **Christian Koch**: Formal analysis; Validation; Investigation; Visualization; Writing—original draft; Writing—review and editing. **Florin Schneiter**: Formal analysis; Investigation; Writing—review and editing. **Jonas Fullin**: Formal analysis; Investigation; Writing—review and editing. **Veronika Lysenko**: Formal analysis; Investigation; Writing—review and editing. **Christian Pellegrino**: Formal analysis; Investigation; Writing—original draft; Writing—review and editing. **Nancy Klemm**: Formal analysis; Investigation; Writing—review and editing. **Norman Russkamp**: Formal analysis; Investigation; Writing—review and editing. **Renier Myburgh**: Formal analysis; Investigation; Writing—review and editing. **Laura Volta**: Formal analysis; Investigation; Writing—review and editing. **Alexandre PA Theocharides**: Conceptualization; Supervision; Investigation; Writing—review and editing. **Kari J Kurppa**: Conceptualization; Formal analysis; Supervision; Writing—original draft; Writing—review and editing. **Benjamin L Ebert**: Conceptualization; Writing—original draft; Writing—review and editing. **Timm Schroeder**: Conceptualization; Data curation; Supervision; Methodology; Writing—original draft; Writing—review and editing. **Markus G Manz**: Conceptualization; Resources; Data curation; Formal analysis; Supervision; Funding acquisition; Writing—original draft; Project administration; Writing—review and editing. **Steffen Boettcher**: Conceptualization; Resources; Data curation; Formal analysis; Supervision; Funding acquisition; Visualization; Methodology; Writing—original draft; Project administration; Writing—review and editing.

## Disclosure and competing interests statement

The authors declare no competing interests.

# Expanded View Figures

**Figure EV1. Details of TP53-associated resistance to in vitro CAR T-cell killing, relates to Figs. 1 and 2.**

(A) Vector design for second-generation CAR expression under an EF1 promotor with a CD8α hinge, a 4-1BB costimulatory domain, a CD3ζ activating domain and an RQR8 identification and selection peptide. (B) Lentiviral T-cell transduction and MACS purification yielded a > 95% pure CAR T-cell population for further experiments. (C) In vitro growth kinetics of unperturbed MOLM13-$TP53^{+/+}$ and MOLM13-$TP53^{-/-}$ leukemia cells. (D) Graphical representation of in vitro competitive co-incubation assay. (E) Representative FACS plots of in vitro killing assays showing ratio of MOLM13-$TP53^{-/-}$GFP$^+$ over MOLM13-$TP53^{+/+}$RFP$^+$ upon 10 days of co-incubation with untransduced T-cell controls and or CAR T-cells. Percentages of parental populations are shown. (F) −Log of MOLM13-$TP53^{+/+}$/MOLM13-$TP53^{-/-}$ ratios normalized to the calculated initial MOLM13-$TP53^{+/+}$/MOLM13-$TP53^{-/-}$ ratios plotted against time of co-incubation. Pooled results of all different E:T and KO:WT ratios are shown (biological replicates, $n = 2$; three technical replicates per biological replicate; symbols represent means and error bars indicate SD; two-way ANOVA). (G) Calculated specific killing from in vitro competitive co-incubation assays of anti-CD33 CAR T-cells against a mixture of MOLM13-$TP53^{+/+}$ and MOLM13-$TP53^{-/-}$ leukemia cells. (H) Absolute CD3$^+$ cell numbers at an E:T ratio of 1:16 for untransduced T-cell controls and anti-CD33 CAR T-cells co-incubated with MOLM13-$TP53^{+/+}$ (black), MOLM13-$TP53^{-/-}$ (red) or MOLM13-$TP53^{missense/-}$ (blue) AML cells on day 6 (biological replicates, $n = 2$; two technical replicates per biological replicate; symbols represent individual replicates; thickened lines indicate means and error bars indicate SD; two-way ANOVA). (I) CD33 target antigen density on MOLM13-$TP53^{+/+}$ and $TP53^{-/-}$ AML cells in co-incubation at an E:T of 1:16 with untransduced T-cell controls and anti-CD33-directed CAR T-cells on days 1 and 6 (biological replicates, $n = 4$; three technical replicates for each biological replicates; symbols represent individual replicates; thickened lines indicate means and error bars indicate SD; two-way ANOVA). (J) PD-L1 surface expression on MOLM13-$TP53^{+/+}$ and $TP53^{-/-}$ AML cells in co-incubation with untransduced T-cell controls and anti-CD33-directed CAR T-cells on days 1 and 6 (biological replicates, $n = 3$; two technical replicates for each biological replicate; symbols represent individual replicates; thickened lines indicate means and error bars indicate SD; two-way ANOVA).

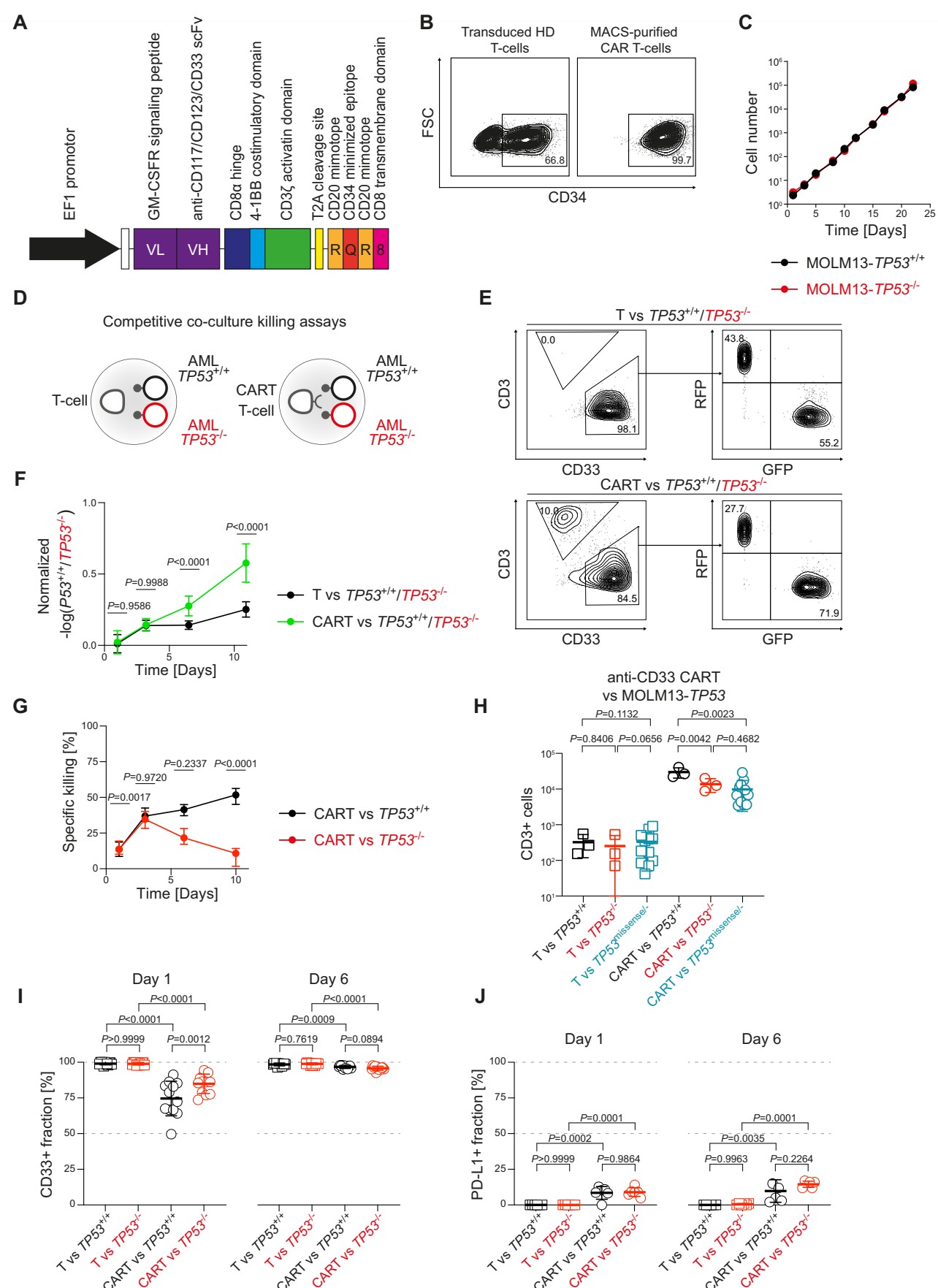

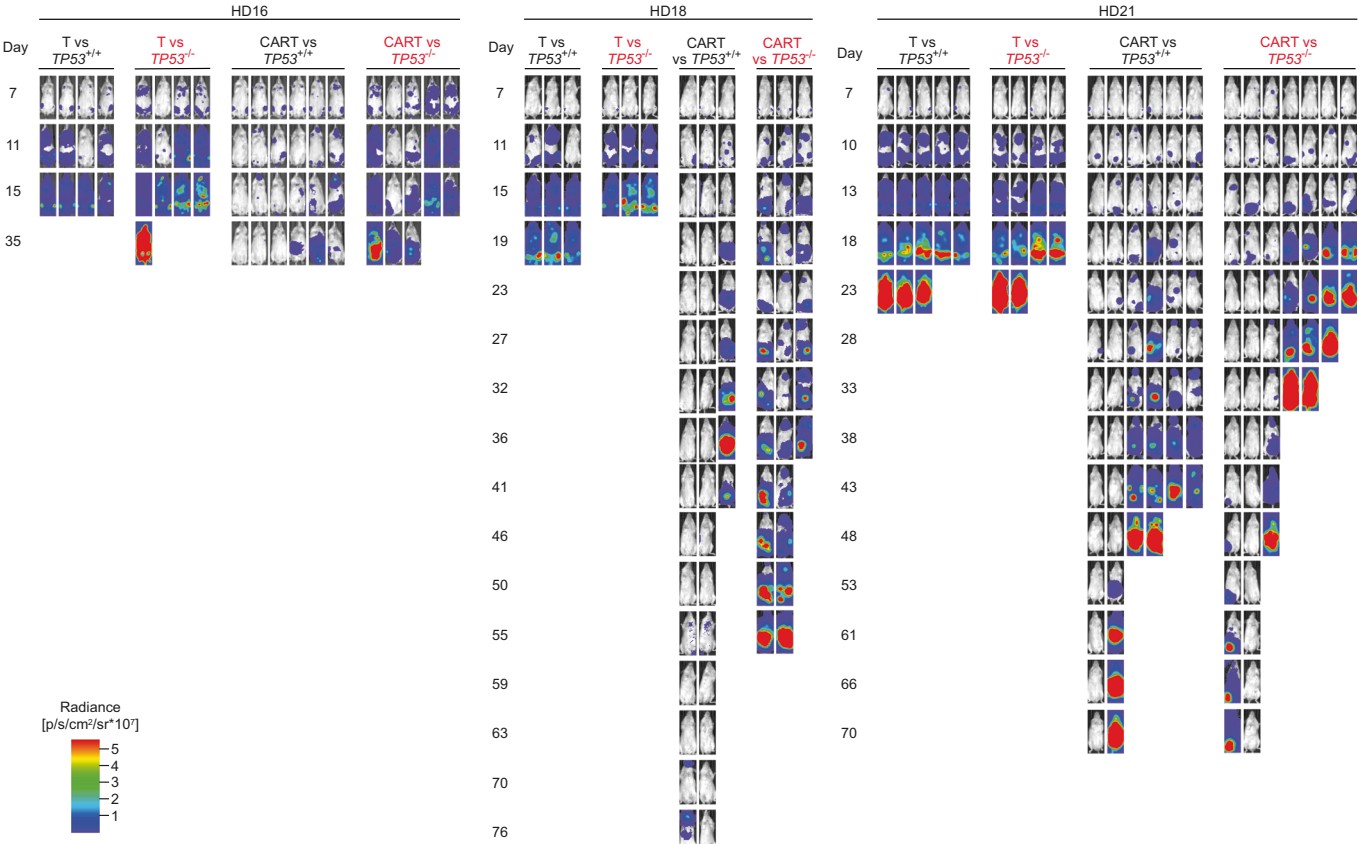

**Figure EV2. Details of therapeutic in vivo xenograft model, relates to Fig. 3.**

Pseudo-colored bioluminescence measurements taken at the indicated days showing leukemic burden in the respective groups of treated and control mice. The three biological replicates with different HD untransduced T-cells and CAR T-cells are shown, ($n = 53$ mice in total and $n = 3$ biological replicates; T-cells vs. MOLM13-$TP53^{+/+}$Luc$^+$=12, T-cells vs. MOLM13-$TP53^{-/-}$Luc$^+$=11, CAR T-cells vs. MOLM13-$TP53^{+/+}$Luc$^+$=15, CAR T-cells vs. MOLM13-$TP53^{-/-}$Luc$^+$=15).

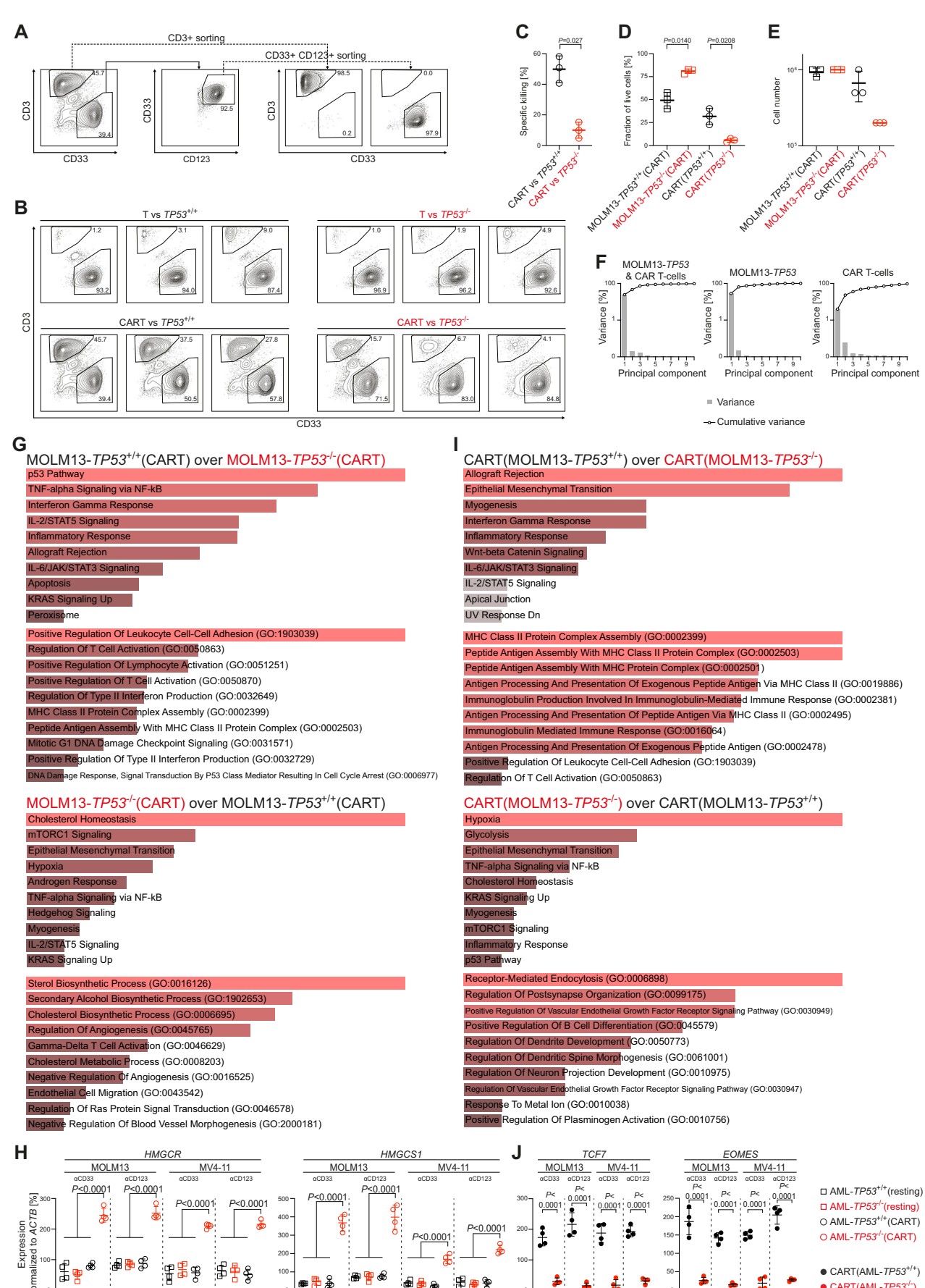

**Figure EV3.  Details of sorting procedure and gene expression profiling results, relates to Fig. 4.**

(A) Gating strategy for flow cytometry sorting as shown by an example (i.e., same data) from panel (B): CD3$^+$ T-cells were sorted as shown, CD3$^-$CD33$^+$ events were further gated on CD33$^+$CD123$^+$ to sort double-positive MOLM13-*TP53* AML leukemia cells. The resulting cell populations of >97% purity are shown on the right two panels. (B) FACS plots of $n = 3$ technical replicates of in vitro co-culture assays subjected to flow cytometry sorting and mRNA extraction. (C) Calculated specific killing from co-incubation assay of MOLM13-*TP53*$^{+/+}$ (black) or MOLM13-*TP53*$^{-/-}$ (red) with anti-CD33 CAR T-cells used for RNA-sequencing (biological replicate, $n = 1$; three technical replicates; symbols indicate individual technical replicates; thickened line represents mean and error bars indicate SD; unpaired Student's $t$ test). (D) Fraction of live cells from co-incubation assay used for RNA-sequencing. (biological repliactes, $n = 1$; 3 technical replicates; symbols represent individual replicates; thickened lines indicate mean and error bars represent SD; unpaired Student's $t$ test). (E) Total sorted cell numbers for the respective conditions used for mRNA sequencing. (F) Proportion of variance plot. Bars represent the specific proportion of total variance explained by the principal component (PC) and curve represents the cumulative variance explained by PC and all PCs before it. (G) Pathways and gene ontology (GO) terms related to biological processes (BPs) enriched in MOLM13-*TP53* under CAR T-cell attack. (H) RT-qPCR measuring expression of *HMGCR* and *HMGCS1* transcripts normalized to *ACTB* in MOLM13-*TP53* and MV4-11-*TP53* cells sorted from co-incubation assays with either untransduced T-cells [AML-*TP53*(resting)] or CD33-/CD123-directed CAR T-cells [AML-*TP53*(CART)] (biological replicates, $n = 2$; 2 technical replicates per biological replicate; symbols indicate individual replicates; thickened lines indicate mean and error bars indicate SD; one-way ANOVA). (I) Pathways and gene ontology (GO) terms related to biological processes (BPs) enriched in CAR T-cells co-incubated with MOLM13-*TP53*. (J) RT-qPCR measuring expression of *TCF7* and *EOMES* transcripts normalized to *ACTB* in CAR T-cells sorted from co-incubation assays with *TP53*$^{+/+}$ [CART(AML-*TP53*$^{+/+}$)] or *TP53*$^{-/-}$ [CART(AML-*TP53*$^{-/-}$)] target cells lines MOLM13 and MV4-11 (biological replicates, $n = 2$; 2 technical replicates per biological replicate; symbols indicate individual replicates; thickened lines indicate mean and error bars indicate SD; unpaired Student's $t$ test).

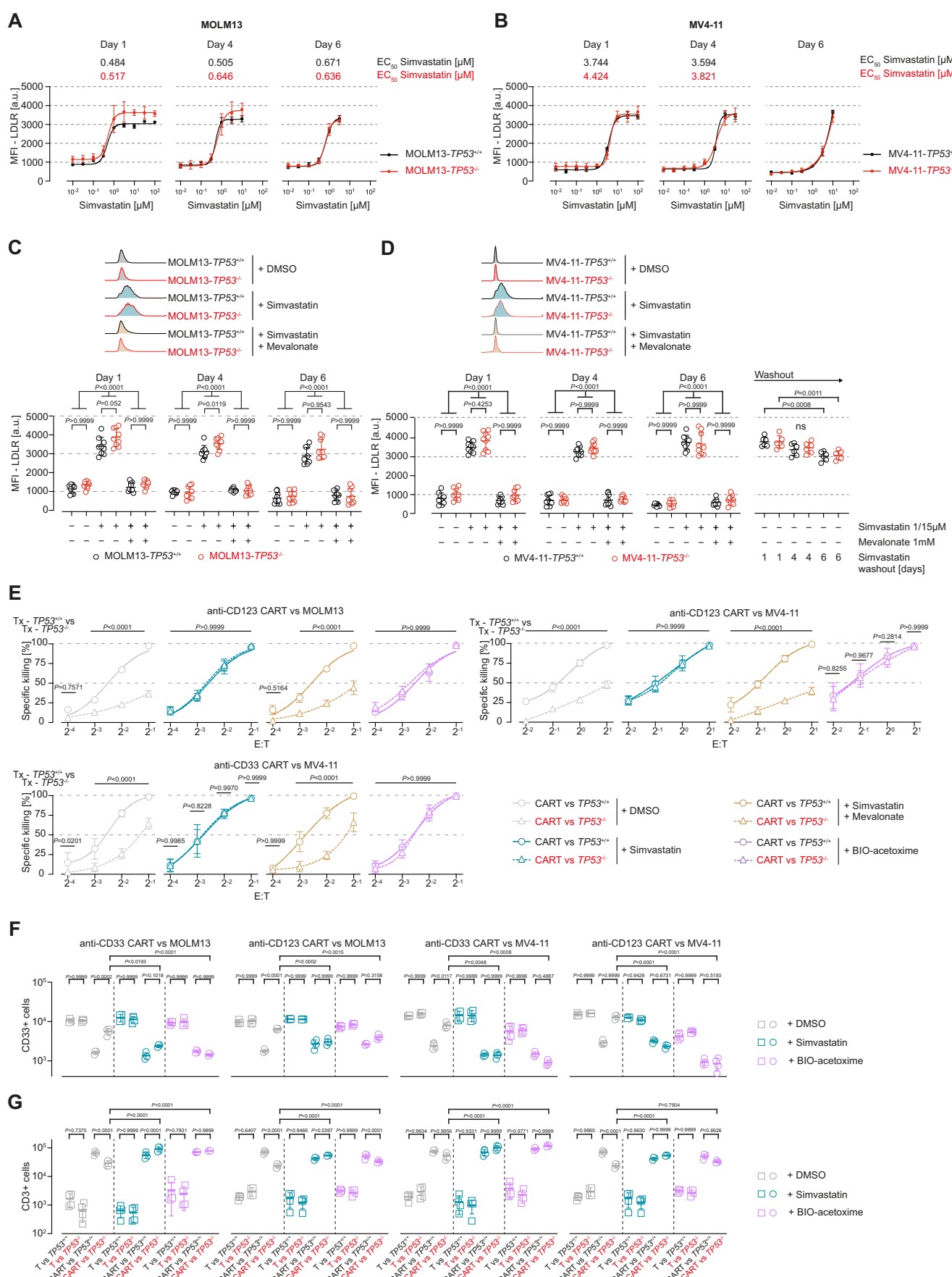

◀  **Figure EV4. Details of rescue co-incubation assays, relates to Figs. 5 and 6.**

LDLR mean fluorescence intensity and extrapolated EC50 of (**A**) MOLM13-*TP53* and (**B**) MV4-11-*TP53* AML cells incubated with increasing concentrations of simvastatin (biological replicates, $n = 2$; symbols represent means; error bars indicate SD). (**C**) Upper panel: Representative FACS histograms of LDLR mean fluorescence intensity of isogenic MOLM13-*TP53* AML cells with wild-type (MOLM13-*TP53*$^{+/+}$) or null (MOLM13-*TP53*$^{-/-}$) *TP53* status in the presence of DMSO, simvastatin 1 μM or simvastatin 1 μM + mevalonate 1mM. Lower panel: LDLR mean fluorescence intensity of MOLM13-*TP53* cells on days 1, 4 and 6 in the presence or absence of simvastatin 1 μM and/or mevalonate 1mM (biological replicates, $n = 3$; 2 technical replicates per biological replicate; symbols indicate individual replicates; thickened lines indicate means and error bars indicate SD; two-way ANOVA). (**D**) Upper panel: Representative FACS histograms of LDLR mean fluorescence intensity of isogenic MV4-11-*TP53* AML cells with wild-type (MV4-11-*TP53*$^{+/+}$) or null (MV4-11-*TP53*$^{-/-}$) *TP53* status in the presence of DMSO, simvastatin 15 μM or simvastatin 15 μM + mevalonate 1mM. Lower panel: LDLR mean fluorescence intensity of MV4-11-*TP53* cells on days 1, 4 and 6 in the presence or absence of simvastatin 15 μM and/or mevalonate 1mM as well as after washout of simvastatin (biological replicates, $n = 3$; 2 technical replicates per biological replicate; symbols indicate individual replicates; thickened lines indicate means and error bars indicate SD; two-way ANOVA). (**E**) Summary data showing results from 3 different co-incubation assays of CAR T-cell-mediated killing of *TP53*$^{+/+}$ or *TP53*$^{-/-}$ AML cells over various E:T ratios in the presence of DMSO, simvastatin (1 μM for MOLM13 and 15 μM for MV4-11), simvastatin + mevalonate 1mM or BIO-acetoxime 0.5 μM, respectively. Co-incubation assays include anti-CD123 CAR vs MOLM13-*TP53*, anti-CD33 CAR vs MV4-11-*TP53* and anti-CD123 CAR vs MV4-11-*TP53*. (replicates, $n = 2$–3; symbols represent mean; error bars indicate SD; two-way ANOVA). (**F**) Absolute CD33+ target cell numbers from various co-incubation assays of untransduced T-cells or CAR T-cells with *TP53*$^{+/+}$ or *TP53*$^{-/-}$ AML cells at an E:T of 1:16 in the presence of DMSO, simvastatin (1 μM for MOLM13 and 15 μM for MV4-11) or BIO-acetoxime 0.5 μM, respectively. Co-incubation assays include anti-CD33 CAR vs MOLM13-*TP53*, anti-CD123 CAR vs MOLM13-*TP53*, anti-CD33 CAR vs MV4-11-*TP53* and anti-CD123 CAR vs MV4-11-*TP53*. (biological replicates, $n = 2$; 2 technical replicates per biological replicate; symbols represent individual replicates; thickened lines indicate means and error bars indicate SD; ns, non-significant; two-way ANOVA). (**G**) Absolute CD3+ effector cell numbers from various co-incubation assays of untransduced T-cells or CAR T-cells with *TP53*$^{+/+}$ or *TP53*$^{-/-}$ AML cells at an E:T of 1:16 in the presence of DMSO, simvastatin (1 μM for MOLM13 and 15 μM for MV4-11) or BIO-acetoxime 0.5 μM, respectively. Co-incubation assays include anti-CD33 CAR vs MOLM13-*TP53*, anti-CD123 CAR vs MOLM13-*TP53*, anti-CD33 CAR vs MV4-11-*TP53* and anti-CD123 CAR vs MV4-11-*TP53*. (biological replicates, $n = 2$; 2 technical replicates per biological replicate; symbols represent individual replicates; thickened lines indicate means and error bars indicate SD; two-way ANOVA).

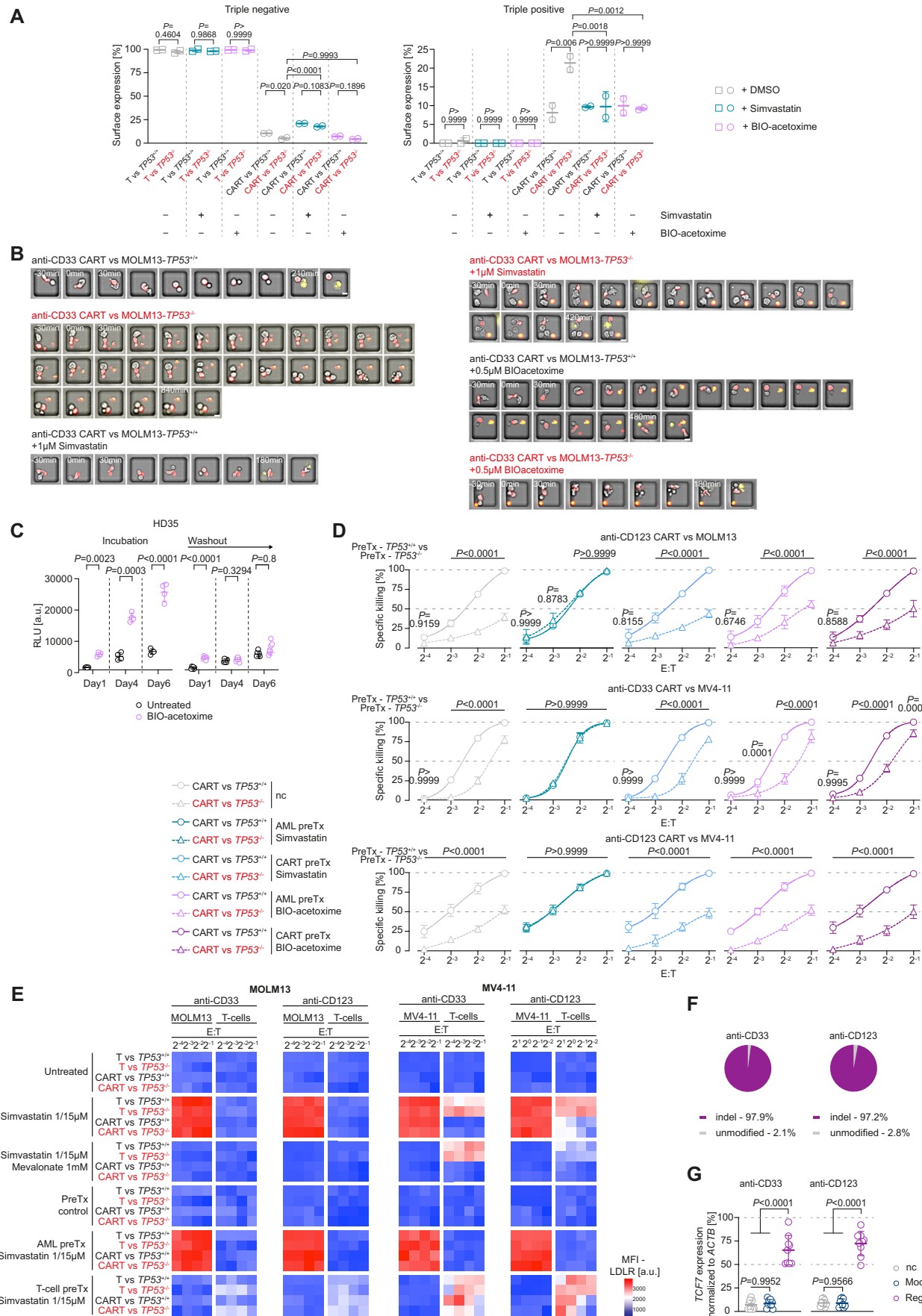

◀ **Figure EV5. Details of rescue and pretreatment co-incubation assays as well as Regnase-1-deficient CAR T-cells, relates to Figs. 5 and 6.**

(A) Fraction of T-cells negative for all three investigated exhaustion markers (right) and negative for all three markers (left) treated with the indicated compounds (one biological replicate, $n = 2$ technical replicates; symbols represent means; error bars indicate SD; two-way ANOVA). (B) Representative stills of fluorescence live-cell time-lapse imaging data of anti-CD33 CAR T-cell engaging MOLM13-$TP53^{+/+}$ AML or MOLM13-$TP53^{-/-}$ AML cells in the presence of simvastatin 1 µM or BIO-acetoxime 0.5 µM (scale bars, 10µm). (C) Luminescence signal of T-cells transduced with a Wnt-responsive luciferase gene reporter system and incubated with or without BIO-acetoxime at 0.5 µM for 1, 4 or 6 days as well as after washout of BIO-acetoxime. (Data shown from a second healthy T-cell donor than in Fig. 6C; biological replicates, $n = 2$; 2 technical replicates per biological replicate; symbols indicate individual replicates; thickened lines denote means and error bars indicate SD; unpaired Student's $t$ test). (D) Results from 3 different pretreatment co-incubation assays with anti-CD33 and anti-CD123 CAR T-cells against $TP53^{+/+}$ or $TP53^{-/-}$ AML (MOLM13 and MV4-11) cells at various E:T ratios. Co-incubations include: anti-CD123 CAR vs MOLM13-$TP53$, anti-CD33 CAR vs MV4-11-$TP53$ and anti-CD33 CAR vs MV4-11-$TP53$. (biological replicates, $n = 2$–3; 2 technical replicates per biological replicate; symbols represent means; error bars indicate SD; two-way ANOVA). (E) Heatmaps depicting LDLR expression (mean fluorescence intensity) within treatment as well as pretreatment co-incubation assays with anti-CD33 and anti-CD123 CAR T-cells against MOLM13-$TP53$ and MV4-11-$TP53$ AML cells at various E:T ratios. (biological replicates, $n = 3$–4; 2 technical replicates per biological replicate; pseudocolors indicate signal of LDLR expression). (F) Sequencing results graphed as %indels of Regnase-1 CRISPRed anti-CD33 as well as anti-CD123 CAR T-cells from one healthy donor. (G) RT-qPCR measuring expression of $TCF7$ transcripts normalized to $ACTB$ in nc, mock and RegKO anti-CD33 and anti-CD123 CAR T-cells (biological replicates, $n = 2$; 2 technical replicates per biological replicate; symbols indicate individual replicates; thickened lines indicate means and error bars indicate SD; paired Student's $t$ test).

