## [Peer Review File · EMBO Molecular Medicine]

Targeting the mevalonate or Wnt pathways to overcome CAR T-cell resistance in TP53-mutant AML cells

Steffen Boettcher, Jan Mueller, Roman Schimmer, Christian Koch, Florin Schneiter, Jonas Fullin, Veronika Lysenko, Christian Pellegrino, Nancy Klemm, Norman Russkamp, Renier Myburgh, Laura Volta, Alexandre Theocharides, Kari Kurppa, Benjamin Ebert, Timm Schroeder, and Markus G. Manz

DOI: [10.15252/emmm.202317767](https://doi.org/10.15252/emmm.202317767)

Corresponding authors: Steffen Boettcher (steffen.boettcher@usz.ch) , Markus G. Manz (markus.manz@usz.ch)

Review Timeline:

Submission Date:	27th Mar 23
Editorial Decision:	19th Apr 23
Revision Received:	28th Nov 23
Editorial Decision:	18th Dec 23
Revision Received:	4th Jan 24
Accepted:	8th Jan 24

Editor: Lise Roth

Transaction Report:

19th Apr 2023

Dear Prof. Boettcher,

Thank you for the submission of your manuscript to EMBO Molecular Medicine. We have now received feedback from the reviewers who agreed to evaluate your manuscript. As you will see from the reports below, the referees acknowledge the interest of the study and are overall supporting publication of your work pending appropriate revisions.

As the requested revisions appear to require a lot of experimental work, I further cross-commented with the referees. Concerning the extension of your findings to primary cells, they agreed that given the difficulty to perform these experiments, validation of key in vitro experiments with other isogenic cell line, MV4-11, and/or some of the missense TP53 mutants, would be sufficient. Please highlight in the discussion that this is a limited model that may not be generalizable to other diseases.

Further validation of RNAseq data, looking at the effects of the inhibitors in each pathway, and normalization of interaction times should be addressed.

As revising the manuscript according to the referees' recommendations appears to require a lot of additional work and experimentation, and given the potential interest of your findings, we are ready to extend the deadline from 3 to 6 months with the understanding that acceptance of the manuscript would entail a second round of review.

EMBO Molecular Medicine encourages a single round of revision only and therefore, acceptance or rejection of the manuscript will depend on the completeness of your responses included in the next, final version of the manuscript. For this reason, and to save you from any frustrations in the end, I would strongly advise against returning an incomplete revision. Should you find that the requested revisions are not feasible within the constraints outlined here and prefer, therefore, to submit your paper elsewhere, we would welcome a message to this effect.

We require:

- 1) A .docx formatted version of the manuscript text (including legends for main figures, EV figures and tables). Please make sure that the changes are highlighted to be clearly visible.
- 2) Individual production quality figure files as .eps, .tif, .jpg (one file per figure). For guidance, download the 'Figure Guide PDF' (<https://www.embopress.org/page/journal/17574684/authorguide#figureformat>).
- 3) At EMBO Press we ask authors to provide source data for the main figures. Our source data coordinator will contact you to discuss which figure panels we would need source data for and will also provide you with helpful tips on how to upload and organize the files.
- 4) A .docx formatted letter INCLUDING the reviewers' reports and your detailed point-by-point responses to their comments. As part of the EMBO Press transparent editorial process, the point-by-point response is part of the Review Process File (RPF), which will be published alongside your paper.
- 5) A complete author checklist, which you can download from our author guidelines (<https://www.embopress.org/page/journal/17574684/authorguide#submissionofrevisions>). Please insert information in the checklist that is also reflected in the manuscript. The completed author checklist will also be part of the RPF.
- 6) Please note that all corresponding authors are required to supply an ORCID ID for their name upon submission of a revised manuscript.
- 7) It is mandatory to include a 'Data Availability' section after the Materials and Methods. Before submitting your revision, primary datasets produced in this study need to be deposited in an appropriate public database, and the accession numbers and database listed under 'Data Availability'. Please remember to provide a reviewer password if the datasets are not yet public (see <https://www.embopress.org/page/journal/17574684/authorguide#dataavailability>).

In case you have no data that requires deposition in a public database, please state so in this section (This study includes no data deposited in external repositories). Note that the Data Availability Section is restricted to new primary data that are part of this study.

- 8) For data quantification: please specify the name of the statistical test used to generate error bars and P values, the number (n) of independent experiments (specify technical or biological replicates) underlying each data point and the test used to

calculate p-values in each figure legend. The figure legends should contain a basic description of n, P and the test applied. Graphs must include a description of the bars and the error bars (s.d., s.e.m.). Please provide exact p values.

13) Author contributions: CRediT has replaced the traditional author contributions section because it offers a systematic machine readable author contributions format that allows for more effective research assessment. Please remove the Authors Contributions from the manuscript and use the free text boxes beneath each contributing author's name in our system to add specific details on the author's contribution. More information is available in our guide to authors.

16) As part of the EMBO Publications transparent editorial process initiative (see our Editorial at <http://embomolmed.embopress.org/content/2/9/329>), EMBO Molecular Medicine will publish online a Review Process File (RPF) to accompany accepted manuscripts.

In the event of acceptance, this file will be published in conjunction with your paper and will include the anonymous referee reports, your point-by-point response and all pertinent correspondence relating to the manuscript. Let us know whether you agree with the publication of the RPF and as here, if you want to remove or not any figures from it prior to publication. Please note that the Authors checklist will be published at the end of the RPF.

EMBO Molecular Medicine has a "scooping protection" policy, whereby similar findings that are published by others during review or revision are not a criterion for rejection. Should you decide to submit a revised version, I do ask that you get in touch

after three months if you have not completed it, to update us on the status.

I look forward to receiving your revised manuscript.

Yours sincerely,

Lise Roth

***** Reviewer's comments *****

Referee #1 (Comments on Novelty/Model System for Author):

All results are obtained in a single cell line model. Although this model is adequate, the general relevance of the concepts presented in this work should be addressed by including more cell line models as well as primary patient-derived cells.

Referee #1 (Remarks for Author):

The study by Mueller et al. uses isogenic variants of the MOLM-13 cell line to show that loss of the tumor suppressor gene TP53 cause resistance to CAR T-cell-mediated killing. While most of the work uses CD33-directed CAR T-cells and TP53^{-/-} MOLM-13 cells, they also show that similar effects are observed with CAR-T cells targeting another surface molecule expressed on MOLM-13 cells (such as CD123). Also, missense mutations in TP53 caused a similar degree of resistance to CAR T-cell mediated killing as TP53 loss. Upon incubation with TP53^{-/-} cells, CAR T-cells exhibited an immunophenotype that was associated with immune exhaustion and increased trogocytosis. Live imaging showed that TP53-deficient MOLM-13 cells exhibit extended interactions with CAR T-cells. Resistance of TP53^{-/-} MOLM13 cells to CAR T-cell mediated killing was also observed in vivo in a xenograft mouse model. Gene expression profiling revealed that loss of TP53 caused upregulated expression of genes involved in cholesterol biosynthesis in CAR T-cell exposed MOLM-13 cells, while CAR T-cells exposed to TP53-deficient MOLM-13 cells showed increased expression of the Wnt target genes TCF7 and EOMES. Pharmacologic inhibition of cholesterol biosynthesis via Simvastatin as well as activation of the WNT signaling pathway through BIOacetoxime restored the antileukemic activity of CD33-directed CAR T-cells against TP53^{-/-} MOLM13 cells in co culture assays.

This is an interesting manuscript that presents interesting findings that could have translational potential for CAR T-cell based therapies in TP53-mutated AML. The manuscript is clearly written and the results are presented in a logical order. However, there are several shortcomings that have to be addressed before the work can be accepted for publication.

Main points:

- All results presented in this manuscript were obtained in a single cell line model. The authors have to demonstrate the general relevance of their findings in different AML cell lines as well as in primary patient-derived AML cells with different TP53 genotypes. Are TP53-deficient AML cell lines (e.g. THP1, KG-1, HL-60) and primary TP53-deficient AML cells also resistant to a-CD33/a-CD123/a-CD117 CAR T-cells?
- Results of the RNA-seq experiments need to be validated through orthogonal approaches. Can the upregulation of cholesterol biosynthesis in TP53-deficient cells be confirmed by qPCR/Western blotting/enzymatic activity? Is the same effect observed in TP53^{-/-} MOLM-13 cells that were not exposed to CAR T-cells? Is the same effect observed in other TP53-deficient cell lines/primary cells?
- The same applies to the upregulation of Wnt-target genes in CAR T-cells. Can the upregulation be confirmed by qPCR/WB? Is the same effect observed when CAR T-cells are exposed to TP53-deficient cell lines/primary cells? is the same effect observed with a-CD123/a-CD117 CAR T-cells?
- The authors need to provide evidence that the the cholesterol synthesis /Wnt pathways are indeed inhibited at the drug concentrations that were applied to the co-cultures.
- Does inhibition of cholesterol biosynthesis/activation of Wnt signalling restore CAR T-cell activity against other cell line models of TP53-deficient AML/primary AML cells with TP53 mutations?

- As drugs were applied to co-cultures of MOLM-13 and CAR T-cells, it is possible that inhibition of cholesterol synthesis affects CAR T-cells and activation of Wnt signaling affects AML cells. This possibility should be out by pre-treatment of either AML or CAR T-cell populations with the respective inhibitors prior to coculturing them.
- Does inhibition of cholesterol biosynthesis normalize interaction times between TP53^{-/-}-AML cells and CAR T-cells?

Minor points

- Can the authors provide the results of a functional annotation of differentially expressed genes between CAR T-exposed wild type vs TP53^{-/-} MOLM 13 cells by Gene Ontology (GO) or a similar algorithm? Are there other pathways that are enriched/depleted in response to TP53 loss?
- Are the differences between TP53^{+/+} and TP53^{-/-} cells in Figure 6C statistically significant?

Referee #2 (Comments on Novelty/Model System for Author):

The model of using an isogenic cell line is limited due to the artificial nature of cells in culture; however this is the best model available currently.

Referee #2 (Remarks for Author):

Mueller et al show that TP53 null AML cell lines are more resistant to CAR T cell mediated killing compared to isogenic TP53 intact cells. Specifically, while both cell lines are susceptible to CAR T cells at high effector-to-target ratios, TP53 null cells exhibit earlier outgrowth of tumor cells at dose-limiting CAR T cell numbers (~1:16 E:T ratio). The authors show that this is due to prolonged contact between TP53 null targets and CAR T cells leading to increased CAR T cell exhaustion. Furthermore, these effects can be ameliorated by inhibition of cholesterol synthesis in tumor cells or by activating the Wnt pathway in CAR T cells.

This is a novel finding that has broad implications for CAR T cell therapy in general, as so far there are few pre-treatment determinants of resistance other than absence of target antigen. However, I am concerned that these results may be subject to overinterpretation. Specifically, my concerns are as follows:

1. Does TP53 deficiency leads to resistance to CAR T cells in general, or is this limited to AML? TP53 is a commonly mutated gene in a variety of cancers, and it would be informative to know whether TP53 loss predicts poor response to CAR T cells across all tumors. Particularly this would be relevant to B cell ALL, lymphoma and multiple myeloma, diseases in which CAR T cells are already being used in patients. The authors cite clinical studies in lymphoma reporting conflicting effects of TP53 deficiency, but this paper does not actually answer this conundrum.
2. Figure 6B shows that simvastatin and BIOacetoxime improves killing of TP53 null tumor when depicted as %specific killing. However, Fig S8 shows that the absolute numbers of CD33⁺ tumor cells and CD3⁺ T cells are not significantly changed with these drugs. Rather the effect is mediated by increased numbers of CD33⁺ tumor cells in the TP53-null Molm13. It is also notable that in this experiment there is no decrease of CAR T cell numbers seen on incubation with TP53-null Molm13, in contrast to Figure 1F. These discrepancies are concerning in regard to the validity and reproducibility of the results.
3. Have the authors done the experiments in Figure 6 with lower doses of simvastatin?
4. It is unclear how TP53 deficiency leads to increased cholesterol biosynthesis which induces tumor resistance and T cell exhaustion. A more thoughtful discussion of the interplay of these mechanisms would be helpful for readers to digest these findings.

Minor comments:

1. Figure S1 contains key data that should be moved to the main figures if possible, specifically S1D, G, H, I.
2. Figure S1J is difficult to interpret, it would be better to show the ratio of GFP/RFP cells at the end of culture (day 10) as dot plots.
3. Figure 5C is also difficult to interpret due to the multiple lines, which are not necessary as the individual BLI values are already shown in Fig 5B. It would be better to present the data in aggregate (median + error bars).

4. It would be helpful for the figure legends to indicate what the error bars indicate.
5. In Figure 1 it is unclear what "n=3 biological and 7 technical replicates" means, this should be clarified. Does this mean 3 different donors, each done in 7 replicates? Or does this mean that each donor was assessed with 2-3 replicates? The figure suggests the latter, but which data points are technical replicates vs. biologic replicates should be clearly demarcated.
6. Similarly, in Figure 4 "n=53 mice and n=3 healthy T cell donors" is insufficient - the legend should clarify how many mice per each group.
7. Reference #36 by Singh et al - in this CRISPR screen TP53 was not enriched in CART-resistant ALL cells, and p53 signaling was the 6th enriched pathway identified, after apoptosis, ribosome, pathways in cancer, RNA degradation, and RIG1-like receptor signaling. Do the authors have any thoughts on why the TP53 signal is so weak in this model?

Point-by-point response

We would like to thank the editor and reviewers for their time and effort spent on reviewing our manuscript. We feel that the constructive and supportive suggestions helped us substantially improve our study. We are happy to re-submit the revised manuscript and hope that, after having addressed all the comments and suggestions, our manuscript will now prove satisfactory for acceptance for publication.

Dear Prof. Boettcher,

Thank you for the submission of your manuscript to EMBO Molecular Medicine. We have now received feedback from the reviewers who agreed to evaluate your manuscript. As you will see from the reports below, the referees acknowledge the interest of the study and are overall supporting publication of your work pending appropriate revisions.

As the requested revisions appear to require a lot of experimental work, I further cross-commented with the referees. Concerning the extension of your findings to primary cells, they agreed that given the difficulty to perform these experiments, **validation of key in vitro experiments with other isogenic cell line, MV4-11, and/or some of the missense TP53 mutants**, would be sufficient. Please highlight in the discussion that this is a limited model that may not be generalizable to other diseases.

Further validation of RNAseq data, looking at the effects of the inhibitors in each pathway, and normalization of interaction times should be addressed.

As revising the manuscript according to the referees' recommendations appears to require a lot of additional work and experimentation, and given the potential interest of your findings, we are ready to extend the deadline from 3 to 6 months with the understanding that acceptance of the manuscript would entail a second round of review.

EMBO Molecular Medicine encourages a single round of revision only and therefore, acceptance or rejection of the manuscript will depend on the completeness of your responses included in the next, final version of the manuscript. For this reason, and to save you from any frustrations in the end, I would strongly advise against returning an incomplete revision. Should you find that the requested revisions are not feasible within the constraints outlined here and prefer, therefore, to submit your paper elsewhere, we would welcome a message to this effect.

We require:

4) A .docx formatted letter INCLUDING the reviewers' reports and your detailed point-by-point responses to their comments. As part of the EMBO Press transparent editorial process, the point-by-point response is part of the Review Process File (RPF), which will be published alongside your paper.

5) A complete author checklist, which you can download from our author guidelines (<https://www.embopress.org/page/journal/17574684/authorguide#submissionofrevisions>). Please insert information in the checklist that is also reflected in the manuscript. The completed author checklist will also be part of the RPF.

6) Please note that all corresponding authors are required to supply an ORCID ID for their name upon submission of a revised manuscript.

7) It is mandatory to include a 'Data Availability' section after the Materials and Methods. Before submitting your revision, primary datasets produced in this study need to be deposited in an appropriate public database, and the accession numbers and database listed under 'Data Availability'. Please remember to provide a reviewer password if the datasets are not yet public (see <https://www.embopress.org/page/journal/17574684/authorguide#dataavailability>).

In case you have no data that requires deposition in a public database, please state so in this section (This study includes no data deposited in external repositories). Note that the Data Availability Section is restricted to new primary data that are part of this study.

8) For data quantification: please specify the name of the statistical test used to generate error bars and P values, the number (n) of independent experiments (specify technical or biological replicates) underlying each data point and the test used to calculate p-values in each figure legend. The figure legends should contain a basic description of n, P and the test applied. Graphs must include a description of the bars and the error bars (s.d., s.e.m.). Please provide exact p values.

9) Our journal encourages inclusion of *data citations in the reference list* to directly cite datasets that were re-used and obtained from public databases. Data citations in the article text are distinct from normal bibliographical citations and should directly link to the database records from which the data can be accessed. In the main text, data citations are formatted as follows: "Data ref: Smith et al, 2001" or "Data ref: NCBI Sequence Read Archive PRJNA342805, 2017". In the Reference list, data citations must be labeled with "[DATASET]". A data reference must provide the database name, accession number/identifiers and a resolvable link to the landing page from which the data can be accessed at the end of the reference. Further instructions are available at <https://www.embopress.org/page/journal/17574684/authorguide#referencesformat>.

<https://www.embopress.org/page/journal/17574684/authorguide#expandedview>

13) Author contributions: CRediT has replaced the traditional author contributions section because it offers a systematic machine readable author contributions format that allows for more effective research assessment. Please remove the Authors Contributions from the

manuscript and use the free text boxes beneath each contributing author's name in our system to add specific details on the author's contribution. More information is available in our guide to authors.

16) As part of the EMBO Publications transparent editorial process initiative (see our Editorial at <http://embomolmed.embopress.org/content/2/9/329>), EMBO Molecular Medicine will publish online a Review Process File (RPF) to accompany accepted manuscripts.

In the event of acceptance, this file will be published in conjunction with your paper and will include the anonymous referee reports, your point-by-point response and all pertinent correspondence relating to the manuscript. Let us know whether you agree with the publication of the RPF and as here, if you want to remove or not any figures from it prior to publication.

I look forward to receiving your revised manuscript.

Yours sincerely,

Lise Roth

Lise Roth, PhD

Senior Editor

EMBO Molecular Medicine

**** Reviewer's comments ****

Referee #1 (Comments on Novelty/Model System for Author):

All results are obtained in a single cell line model. Although this model is adequate, the general relevance of the concepts presented in this work should be addressed by including more cell line models as well as primary patient-derived cells.

Referee #1 (Remarks for Author):

The study by Mueller et al. uses isogenic variants of the MOLM-13 cell line to show that loss of the tumor suppressor gene TP53 cause resistance to CAR T-cell-mediated killing. While most of the work uses CD33-directed CAR T-cells and TP53^{-/-} MOLM-13 cells, they also show that similar effects are observed with CAR-T cells targeting another surface molecule expressed on MOLM-13 cells (such as CD123). Also, missense mutations in TP53 caused a similar degree of resistance to CAR T-cell mediated killing as TP53 loss. Upon incubation with TP53^{-/-} cells, CAR T-cells exhibited an immunophenotype that was associated with immune exhaustion and increased trogocytosis. Live imaging showed that TP53-deficient MOLM-13 cells exhibit extended interactions with CAR T-cells. Resistance of TP53^{-/-} MOLM13 cells to CAR T-cell mediated killing was also observed in vivo in a xenograft mouse model. Gene expression profiling revealed that loss of TP53 caused upregulated expression of genes involved in cholesterol biosynthesis in CAR T-cell exposed MOLM-13 cells, while CAR T-cells exposed to TP53-deficient MOLM-13 cells showed increased expression of the Wnt target genes TCF7 and EOMES. Pharmacologic inhibition of cholesterol biosynthesis via Simvastatin as well as activation of the WNT signaling pathway through BIOacetoxime restored the antileukemic activity of CD33-directed CAR T-cells against TP53^{-/-} MOLM13 cells in co culture assays.

This is an interesting manuscript that presents interesting findings that could have translational potential for CAR T-cell based therapies in TP53-mutated AML. The manuscript is clearly written and the results are presented in a logical order. However, there are several shortcomings that have to be addressed before the work can be accepted for publication.

Main points:

1. All results presented in this manuscript were obtained in a single cell line model. The authors have to demonstrate the general relevance of their findings in different AML cell lines as well as in primary patient-derived AML cells with different TP53 genotypes. Are TP53-deficient AML cell lines (e.g., THP1, KG-1, HL-60) and primary TP53-deficient AML cells also resistant to a-CD33/a-CD123/a-CD117 CAR T-cells?

Response to reviewer 1: The reviewer raises the very important concern that the current study primarily focuses on a single AML cell line model (i.e., MOLM13) and this may compromise the generalizability of our findings. We have therefore performed an extensive set of new experiments with additional isogenic MOLM13 cells carrying the six most common *TP53* missense mutations (R175H^{-/-}, Y220C^{-/-}, M237^{-/-}, R248Q^{-/-}, R273H^{-/-} and R282W^{-/-}). Importantly, we generated a second *TP53* isogenic human AML cell line model based on MV4-11 cells using CRISPR/Cas9 gene editing. With these additional cell lines, we obtained highly robust and clear results corroborating our previous data. We have added the new data to the revised version of the manuscript.

Below is a list of all experiments performed with the additional cell lines as well as their corresponding figure:

- MOLM13-*TP53* (WT, KO, and missense mutations):
 - Co-incubation assays with MOLM13 and anti-CD33 as well as anti-CD123 CAR T-cells. (Fig. 1F and G)
- MV4-11-*TP53* (WT and KO)
 - Co-incubation assays with MV4-11 and anti-CD33 as well as anti-CD123 CAR T-cells.. (Fig. 1I)
 - RNA-seq validation experiments by RT-qPCR with MV4-11 (*TP53*-WT and *TP53*-KO; Fig. EV3H and J)
 - Simvastatin on-target activity experiments with MV4-11 using a luciferase gene reporter assays, LDLR expression experiments as well as mevalonate rescue experiments (Fig. EV4B, D and E, EV5E; Appendix Fig. S5C)
 - Co-incubation assays with simvastatin and BIO-acetoxime with MV4-11 and anti-CD33 as well as anti-CD123 CAR T-cells. (Fig. EV4E-G).
 - Pretreatment co-incubation assays with MV4-11 and anti-CD33 as well as anti-CD123 CAR T-cells. (Fig. EV5D).

We fully agree with the reviewer that experimentation in primary patient cells is desirable whenever feasible. Nevertheless, the use of CRISPR/Cas9-engineered isogenic cell lines provides the unique ability to compare the functional consequences of loss / mutation of a gene of interest (in this case *TP53*) on a biological read-out (in this case response to CAR T-cell therapy) in cells that only differ in the allelic configuration of that gene of interest (i.e., *TP53*) but that are otherwise genetically identical. We believe that our study is a prime example of how this approach can be successfully applied to advance our understanding of disease processes.

2. Results of the RNA-seq experiments need to be validated through orthogonal approaches. Can the upregulation of cholesterol biosynthesis in *TP53*-deficient cells be confirmed by qPCR/Western blotting/enzymatic activity? Is the same effect observed in *TP53*^{-/-} MOLM-13 cells that were not exposed to CAR T-cells? Is the same effect observed in other *TP53*-deficient cell lines/primary cells?

Response to reviewer 1: We followed the reviewer's suggestion to perform validation experiments of the RNA-seq data. To this end, we have investigated *HMGCR* and *HMGCS1* transcript levels by RT-qPCR for both MOLM13-*TP53* as well as MV4-11-*TP53* sorted from co-incubation assays with untransduced T cells as well as anti-CD33 and anti-CD123 CAR T-cells and added the data to Figure EV3H. The results indicate that the upregulation of the mevalonate pathway is robustly observed in both isogenic cell line models, occurs irrespective of the targeted antigen, and is specifically observed in *TP53*-KO cells under CAR T-cell attack.

3. The same applies to the upregulation of Wnt-target genes in CAR T-cells. Can the upregulation be confirmed by qPCR/WB? Is the same effect observed when CAR T-cells are exposed to *TP53*-deficient cell lines/primary cells? Is the same effect observed with a-CD123/a-CD117 CAR T-cells?

Response to reviewer 1: In line with above mentioned experiments, we additionally analyzed transcript levels of *TCF7* and *EOMES* in CAR T-cells sorted from co-incubation assays that were exposed to AML cells with either wildtype or mutated *TP53* (MOLM13

and MV4-11) by RT-qPCR and added the data to Figure EV3J. The results confirm that the upregulation of *TCF7* and *EOMES* in CAR T-cells is robustly observed in both isogenic cell line models, occurs irrespective of the targeted antigen, and is specifically observed in CAR T-cells attacking AML cells with wildtype *TP53*, i.e., associates with an effective CAR T-cell response against the target antigen.

4. The authors need to provide evidence that the cholesterol synthesis /Wnt pathways are indeed inhibited at the drug concentrations that were applied to the co-cultures.

Response to reviewer 1: We thank the reviewer for this excellent suggestion to demonstrate on-target activity of simvastatin as well as BIO-acetoxime at given drug concentrations as applied in the co-culture experiments.

Regarding simvastatin: since direct measurement of enzyme activity of HMGCR is, unfortunately, not easily feasible, we decided to demonstrate on-target activity of simvastatin indirectly on multiple orthogonal levels. First, we demonstrate increased activity of sterol-regulatory element-binding proteins (SREBPs) upon simvastatin treatment in isogenic AML cells (MOLM13-*TP53* and MV4-11-*TP53*) using a luciferase gene reporter system (Appendix Fig. S5C). Second, we demonstrate the resulting increase of low-density lipoprotein-receptor (LDLR) expression upon simvastatin exposure in target cells both in AML cells alone (Fig. EV4A-D) as well as within co-incubation assays (Fig. EV5E). And third, we are able to abrogate the effects of simvastatin in all performed experiments – especially in the co-incubation assays – by adding mevalonate, the direct metabolite produced by HMGCR, to simvastatin-treated cells (Fig. EV4C-E, EV5E, Appendix Fig. S5C). Whilst we cannot show direct enzymatic inhibition of simvastatin on HMGCR, we demonstrate on-target activity of simvastatin using multiple orthogonal approaches.

Regarding BIO-acetoxime: analogous to the abovementioned reporter assay, we demonstrate Wnt pathway activation by BIO-acetoxime using T-cells lentivirally transduced to express luciferase coupled to Wnt-responsive promoters (Fig. 6C, EV5B).

Altogether, we can provide compelling evidence that the mevalonate and Wnt pathways are effectively inhibited and activated at the designated drug concentrations, respectively.

5. Does inhibition of cholesterol biosynthesis/activation of Wnt signalling restore CAR T-cell activity against other cell line models of *TP53*-deficient AML/primary AML cells with *TP53* mutations?

Response to reviewer 1: We performed co-incubation assays with isogenic MV4-11-*TP53* cells with anti-CD33 and anti-CD123 CAR T-cells in the presence of simvastatin or BIO-acetoxime. In this different *TP53*-deficient AML cell line model, we were able to corroborate all our previous findings, namely that the addition of simvastatin or BIO-acetoxime rescues the efficacy of CAR T-cells in killing *TP53*-mutant AML cells (Fig. EV4E-G).

Of note however, we observed from the experiments performed for major revision point 4 that MV4-11 cells are generally less sensitive to simvastatin exposure (Fig. EV4B). Therefore, an increased dose of simvastatin was necessary to observe rescue of the efficacy of CAR T-cells in killing *TP53*-mutant AML cells (Fig. EV4E).

6. As drugs were applied to co-cultures of MOLM-13 and CAR T-cells, it is possible that inhibition of cholesterol synthesis affects CAR T-cells and activation of Wnt signaling

affects AML cells. This possibility should be out by pre-treatment of either AML or CAR T-cell populations with the respective inhibitors prior to coculturing them.

Response to reviewer 1: We are grateful to the reviewer for raising this important point. We have therefore performed co-incubation assays with pre-treated AML and CAR T-cells, respectively. To this end, we pre-treated both AML and CAR T-cells with either Simvastatin and/or BIO-acetoxime for 24 hours, washed out the drugs, co-incubated them at various effector-to-target ratios, and assessed killing efficacy.

We were able to show that pre-treating *TP53*^{-/-} AML cells with simvastatin rescues *TP53* deficiency-associated resistance against CAR T-cell attack, both in MOLM13 as well as MV4-11 AML cell lines (Fig. 6D, E, EV5D), and that this effect is due to lasting on-target activity of simvastatin (Fig. 6B, EV4D, EV5E). Pre-treating *TP53*^{+/+} AML cells or CAR T-cells with simvastatin has no effect on CAR T-cell killing efficacy (Fig. 6B, EV4D, EV5E).

Similarly, we pre-treated both CAR T-cells and AML cells (both isogenic MOLM13 and MV4-11 cell lines) with BIO-acetoxime but did not observe any significant changes in CAR T-cell mediated killing (Fig. 6D, E, EV5D). This is due to the short-lived action of the GSK3-inhibitor as demonstrated by Wnt reporter assays with subsequent BIO-acetoxime washout (Fig. 6C, Fig. EV5C).

However, to ultimately demonstrate that indeed increased Wnt pathway activation in CAR T-cells is able to overcome *TP53* deficiency-associated resistance of AML cells against CAR T-cell attack, we engineered Regnase-1-deficient CAR T-cells using CRISPR/Cas9 gene editing. Regnase-1 is a ribonuclease that degrades *TCF7* mRNA (encoding TCF1 protein) and has been shown to be associated with increased CAR T-cell efficacy. Using a combinatorial T-cell electroporation and CAR transduction protocol, we were able to efficiently knock out Regnase-1 in CAR T-cells (Reg^{-/-} CAR T-cells) as shown by sequencing (Fig. EV5F). This led to a concomitant increase in *TCF7* mRNA (Fig. EV5G), thereby functionally confirming the Wnt pathway activation. Placed in a co-incubation assay (Fig. 6F), Reg^{-/-} CAR-T cells showed increased killing capabilities and were able to overcome *TP53* deficiency-associated resistance of AML cells against CAR T-cell attack (Fig. 6G, H), ultimately demonstrating that indeed increased Wnt pathway activity is responsible for overcoming the resistance phenotype.

7. Does inhibition of cholesterol biosynthesis normalize interaction times between *TP53*^{-/-} AML cells and CAR T-cells?

Response to reviewer 1: We would like to thank the reviewer for suggesting this crucial experiment. We have performed additional live-cell imaging experiments of co-culture assays in the presence of both simvastatin as well as BIO-acetoxime. We found that the addition of simvastatin or BIO-acetoxime resulted in normalization of cellular interaction times between *TP53*-mutant MOLM13 cells and anti-CD33 CAR T-cells (Fig. 5C, EV5B). Unfortunately, whilst we attempted to expand our data set with the additional isogenic *TP53*-mutant MV4-11 cell line, the increased effector-to-target ratio necessary for effective killing of *TP53*-mutant MV4-11 severely impairs the technical setup of live-cell imaging. We were thus not able to generate these data.

Minor points

1. Can the authors provide the results of a functional annotation of differentially expressed genes between CAR T-exposed wild type vs TP53^{-/-} MOLM 13 cells by Gene Ontology (GO) or a similar algorithm? Are there other pathways that are enriched/depleted in response to TP53 loss?

Response to reviewer 1: We are happy to provide the requested data (Fig. EV3G, I, Fig. S5A, B). While other pathways were enriched too, we focused in this study on the two – in our opinion – most striking and promising pathways.

2. Are the differences between TP53^{+/+} and TP53^{-/-} cells in Figure 6C statistically significant?

Response to reviewer 1: We have added a figure panel (Fig. EV5A) to include a statistical evaluation and commented this in the figure legend. The increase in the fraction of cells triple negative for the investigated exhaustion markers upon simvastatin addition and the decrease of triple positive cells upon simvastatin and BIO-acetoxime addition proved statistically significant.

Referee #2 (Comments on Novelty/Model System for Author):

The model of using an isogenic cell line is limited due to the artificial nature of cells in culture; however this is the best model available currently.

Referee #2 (Remarks for Author):

Mueller et al show that TP53 null AML cell lines are more resistant to CAR T cell mediated killing compared to isogenic TP53 intact cells. Specifically, while both cell lines are susceptible to CAR T cells at high effector-to-target ratios, TP53 null cells exhibit earlier outgrowth of tumor cells at dose-limiting CAR T cell numbers (~1:16 E:T ratio). The authors show that this is due to prolonged contact between TP53 null targets and CAR T cells leading to increased CAR T cell exhaustion. Furthermore, these effects can be ameliorated by inhibition of cholesterol synthesis in tumor cells or by activating the Wnt pathway in CAR T cells.

This is a novel finding that has broad implications for CAR T cell therapy in general, as so far there are few pre-treatment determinants of resistance other than absence of target antigen. However, I am concerned that these results may be subject to overinterpretation. Specifically, my concerns are as follows:

1. Does TP53 deficiency leads to resistance to CAR T cells in general, or is this limited to AML? TP53 is a commonly mutated gene in a variety of cancers, and it would be informative to know whether TP53 loss predicts poor response to CAR T cells across all tumors. Particularly this would be relevant to B cell ALL, lymphoma and multiple myeloma, diseases in which CAR T cells are already being used in patients. The authors cite clinical studies in lymphoma reporting conflicting effects of TP53 deficiency, but this paper does not actually answer this conundrum.

Response to reviewer 2: The reviewer raises an important question as to whether our data on the role of TP53 in AML in the context of CAR T-cell therapies can be translated to other cancer types. It is certainly interesting to explore this aspect further in separate future

studies. In line with the editor's suggestion, we restricted this manuscript to the study of the effects of *TP53* mutations on treatment with CAR T-cell therapy in the setting of AML and have highlighted this restriction in the current manuscript version. Expanding our studies to further include acute lymphoblastic leukemia, Non-hodgkin's lymphoma, and multiple myeloma would have been interesting but would have been beyond the scope of this project necessitating the production of novel *TP53*-mutated isogenic cell lines, various CAR T-cells directed against CD19 and BCMA or even GPRC5D and multiple *in vivo* experiments. In the given time frame, this would not have been feasible. We nonetheless hope that with the addition of the other requested experiments (including another *TP53* KO cell line) the revised manuscript will prove satisfactory for publication.

2. Figure 6B shows that simvastatin and BIOacetoxime improves killing of *TP53* null tumor when depicted as %specific killing. However, Fig S8 shows that the absolute numbers of CD33+ tumor cells and CD3+ T cells are not significantly changed with these drugs. Rather the effect is mediated by increased numbers of CD33+ tumor cells in the *TP53*-null Molm13. It is also notable that in this experiment there is no decrease of CAR T cell numbers seen on incubation with *TP53*-null Molm13, in contrast to Figure 1F. These discrepancies are concerning in regard to the validity and reproducibility of the results.

Response to reviewer 2: We are grateful to the reviewer for highlighting this important aspect. We have performed additional experiments co-incubating both isogenic MOLM13-*TP53* and MV4-11-*TP53* cells with anti-CD33 and anti-CD123 CAR T-cells in the presence of either DMSO only, simvastatin or BIO-acetoxime and measured absolute cell counts after 6 days of incubation. While the addition of these drugs has a small but insignificant cytotoxic effect as shown by decreased total cell counts, the effects are uniformly spread to both CAR T cells and AML cells and we see no difference in AML cell counts when comparing *TP53*-WT and *TP53*-KO AML cells. In the setting of increased CAR T-cell killing (at higher effector-to-target ratios) we can see a relative expansion of CD3+ cells whereas on the contrary we see relative expansion of CD33+ cells in the setting of poor CAR T-cell killing (Figure EV4F and G).

3. Have the authors done the experiments in Figure 6 with lower doses of simvastatin?

Response to reviewer 2: Indeed, we have performed co-incubation assays with lower doses of simvastatin but did not incorporate the results in the revised manuscript. Whilst demonstrating on-target activity of simvastatin in this setting, there is differential sensitivity of AML cell lines to statins as highlighted in the answer to revision point 4 by reviewer 1 (Fig. EV4A, B). We could link on-target activity via LDLR expression to rescue of *TP53* deficiency-associated resistance of AML cells against CAR T-cell attack. When performing co-incubation assays in the presence of simvastatin at a dose that does not result in effective inhibition of the mevalonate pathway (i.e., that does not result in LDLR upregulation) we do not observe rescue of *TP53* deficiency-associated resistance of AML cells against CAR T-cell attack. Ultimately, the doses that were chosen for simvastatin represent the lowest dose at which we can observe this rescue.

Please find below a data set demonstrating the above described. MV4-11 were incubated at a simvastatin dose of 1uM, at which no LDLR upregulation can be seen in comparison to the higher dose of 15uM. When performing co-incubation killing assays at this dose (here with the example of anti-CD123 CAR T-cells) we could not observe rescue of *TP53* deficiency-associated CAR T-cell resistance.

4. It is unclear how TP53 deficiency leads to increased cholesterol biosynthesis which induces tumor resistance and T cell exhaustion. A more thoughtful discussion of the interplay of these mechanisms would be helpful for readers to digest these findings.

Response to reviewer 2: We thank the reviewer for this excellent suggestion. In the revised manuscript in the discussion section, we have further emphasized the well-known function of p53 in regulating the mevalonate pathway. How exactly CAR T cell attack is sensed in *TP53*-mutant AML cells, how this signal subsequently activates the mevalonate pathway eventually inducing resistance is, however, unknown. This will be the scope of future studies. Given that and the wealth of other data in our revised manuscript, we have refrained from too much speculation on these currently unknown mechanisms.

Minor comments:

1. Figure S1 contains key data that should be moved to the main figures if possible, specifically S1D, G, H, I.

Response to reviewer 2: In the revised manuscript, we moved parts of previous Figure S1 into the extended view figures (Fig. EV1).

2. Figure S1J is difficult to interpret, it would be better to show the ratio of GFP/RFP cells at the end of culture (day 10) as dot plots.

Response to reviewer 2: We have adapted the figure as suggested and hope that the panel is now easier to interpret. We analyzed this particular dataset in different ways and decided an even more granular presentation might run the risk of confusing readers, while not adding a major point to the paper. We attach a figure showing the preferential killing of *TP53*-proficient cells at every measured E:T and initial T:T ratio, an important finding we tried to present in a summarized way in FigEV1.

3. Figure 5C is also difficult to interpret due to the multiple lines, which are not necessary as the individual BLI values are already shown in Fig 5B. It would be better to present the data in aggregate (median + error bars).

Response to reviewer 2: We thank the reviewer for this suggestion, Fig. 3D summarizes the bioluminescence data now showing mean + SD.

4. It would be helpful for the figure legends to indicate what the error bars indicate.

Response to reviewer 2: We thank the reviewer for bringing this and the issues stated below to our attention. We have adapted the figure legends accordingly.

5. In Figure 1 it is unclear what "n=3 biological and 7 technical replicates" means, this should be clarified. Does this mean 3 different donors, each done in 7 replicates? Or does this mean that each donor was assessed with 2-3 replicates? The figure suggests the latter, but which data points are technical replicates vs. biologic replicates should be clearly demarcated.

Response to reviewer 2: We have clarified this point in the figure legend and additionally adapted this in all further figures accordingly.

6. Similarly, in Figure 4 "n=53 mice and n=3 healthy T cell donors" is insufficient - the legend should clarify how many mice per each group.

Response to reviewer 2: We have supplemented the figure legends to detail how many mice were used per group.

7. Reference #36 by Singh et al - in this CRISPR screen TP53 was not enriched in CART-resistant ALL cells, and p53 signaling was the 6th enriched pathway identified, after apoptosis, ribosome, pathways in cancer, RNA degradation, and RIG1-like receptor signaling. Do the authors have any thoughts on why the TP53 signal is so weak in this model?

Response to reviewer 2: The reviewer raises an excellent point. We thought about this as well but at this point in time, we have no definitive answer. Please find some thoughts and speculation below:

- While CRISPR screens are powerful to identify novel biology in a largely unbiased, discovery-driven manner, they still have limited sensitivity. For instance, the performance of CRISPR screens is heavily dependent on the sgRNA efficacy in terms of knocking out the gene of interest. By contrast, our hypothesis-driven approach focusing on the effect of a single pathway (i.e., p53) in a genetically well-defined isogenic model system has a much higher sensitivity for that particular pathway.
- In general, CRISPR screens are somewhat artificial, and they do not necessarily reflect naturally-occurring mutational processes. For example, the apoptosis pathway was the top scoring hit in the screen by Singh et al. and they validated their screen findings in a patient cohort, and indeed, found that expression levels of death receptors / apoptosis pathway genes separated patients with good or poor outcomes. However, we cannot assume that direct mutational inactivation of genes of the apoptosis pathway – especially death receptors – would also occur and is then selected for in patients treated with anti-CD19 CAR T-cells developing resistance. In fact, we believe that this is excessively unlikely for multiple reasons including the necessity for multiple mutations to occur in the same cell. It is, however, much more likely that a single central regulator of the apoptosis pathway is the target of mutational inactivation (“to kill two birds with one stone”) and then selected for under the appropriate selective pressure (in this case CAR T-cell attack). In fact, p53 is known to regulate the expression of death receptors (PMID: 29149101). It is tempting to speculate that p53 pathway inactivation (either by direct mutational inactivation or indirectly by, for instance, *MDM2* amplification) led to decreased death receptor expression in patients developing CAR T-cell resistance. We therefore regard the excellent study by Singh et al. not as contradictory but rather confirmatory of our findings, and vice versa.
- Of course, it could just be difference between different disease contexts (AML vs. B-ALL).

18th Dec 2023

Dear Prof. Boettcher,

Thank you for submitting your revised manuscript. We have now received the reports from the referees who re-reviewed your manuscript, and as you will see below, they are supportive of publication pending minor revisions. I will therefore be able to accept your manuscript once the following points will be addressed:

1/ Please address the remaining concern from referee #1.

2/ Manuscript text:

- Please remove the yellow highlights, and only keep in track changes mode any new modification.
- Please provide up to 5 keywords.
- The Materials and Methods section should come after the Discussion and include the text currently in the appendix. Please also provide the following information:
 - o Cell culture: indicate whether the cells were tested for mycoplasma contamination and update the checklist accordingly.
 - o Antibodies: please provide dilutions/concentrations.
 - o Mouse models: indicate the housing and husbandry conditions.
 - o Statistics: please include a statement about blinding, even if no blinding was done, randomization, sample size and inclusion/exclusion criteria, and update the checklist accordingly.
- Data Availability section: This section should contain only the links for deposited datasets in public repositories. Please note that the datasets must be public before acceptance of the manuscript.
- Acknowledgements: the funding information provided in the manuscript should also be entered in the submission system.
- Please rename "Conflict of interests Disclosures" to "Disclosure statement and competing interests": We updated our journal's competing interests policy in January 2022 and request authors to consider both actual and perceived competing interests. Please review the policy <https://www.embopress.org/competing-interests> and update your competing interests if necessary.

3/ Figures:

- Please provide exact p values, not a range, in the figures or in their legends, including for ns - non-significant.
- Movie files: please rename to Movie EV1 and Movie EV2; the legends should be zipped to the files and removed from the appendix.
- Appendix: please add a table of content with page numbers, the supplementary methods should be merged with Materials and Methods in the main manuscript file. The Appendix figures should be added to the PDF. Appendix tables should be renamed "Appendix Table S1" etc. Please remove the yellow highlights.
- Figure legends:
 1. Please note that the figure legend style does not comply with the journal guidelines i.e. all the figure legends are in a run-on style.
 2. Please indicate the statistical test used for data analysis in the legends of figures 4e
 3. Please note that the error bars are not defined in the legend of figures 3d
 4. Please note that information related to n is missing in the legend of figures 4c, f
- References to Appendix Table S2 and S3 are missing in the text.
- Please indicate possible figure re-use in the legends (i.e. Figure EV3A and EV3B).

4/ Source Data: please upload the source data as one file per figure.

5/ The paper explained: I introduced minor modifications, please let me know if you agree or amend as you see fit:

Problem

TP53-mutant acute myeloid leukemia / myelodysplastic neoplasms (AML/MDS) are distinct clinicogenomic entities characterized by chemotherapy resistance, high relapse rates, and poor survival. Chimeric antigen receptor (CAR) T-cell therapy is a successful novel cell-based therapy in hemato-oncology and might also be a promising therapeutic option for TP53-mutant AML/MDS. However, the AML-intrinsic determinants of efficacy of CAR T-cell-based therapies are largely unknown.

Results

Our study shows that TP53 mutations in AML cells lead to increased resistance to CAR T-cells in vitro. CAR T-cells co-incubated with target TP53-mutant AML blasts exhibited decreased proliferation and increased exhaustion compared to wildtype TP53 AML blasts. Live-cell imaging revealed longer time-to-killing of TP53-mutant than wildtype TP53 AML cells upon attack by CAR T-cells, which ultimately led to an inability of CAR T-cells to control TP53-mutant cells. Furthermore, immunodeficient mice xenografted with TP53-mutant AML and treated with CAR T-cells exhibited shorter survival compared to mice engrafted with wildtype TP53 AML. Transcriptional profiling of AML cells with either wildtype or mutant TP53 under CAR T-cell attack revealed upregulation of the mevalonate synthesis pathway in TP53-mutant AML cells. Simultaneously, CAR T-cells engaging TP53-mutant AML demonstrated a downregulated Wnt pathway. Rational pharmacological targeting of either of these pathways

rescued TP53-mutant AML cell sensitivity to CAR T-cell-mediated killing. Similarly, CRISPR/Cas9-engineering of CAR T-cells to upregulate Wnt pathway signaling rescued TP53-mutant AML cell sensitivity and led to improved killing efficacy.

Impact

We demonstrate that TP53 deficiency in AML cells confers resistance to CAR T-cell therapy by inducing CAR T-cell dysfunction. We propose a model in which difficult-to-kill TP53-mutant AML cells promote CAR T-cell exhaustion, eventually leading to uncontrolled AML cell outgrowth. We further identify inhibition of the mevalonate pathway as a potential therapeutic vulnerability of TP53-mutant AML cells, and stimulation of the Wnt pathway as a promising avenue to enhance the efficacy of CAR T-cell therapy. Pharmacological co-interventions or genetic engineering of CAR T-cell products may thus be a promising strategy towards more efficacious and tolerable cellular therapies for patients with TP53-mutant AML / MDS.

6/ Synopsis: I introduced minor modifications to your text, please let me know if you agree with the following or amend as you see fit:

TP53 mutations in AML confer resistance to CAR T-cell therapy through exhaustion of CAR T-cells, and dysregulation of the mevalonate and Wnt pathways in AML and CAR T-cells, respectively. Targeting these pathways holds the promise to improve cellular therapies for TP53-mutant myeloid neoplasms.

- TP53 deficiency in AML cells leads to CAR T-cell exhaustion and ultimately AML resistance and outgrowth.
- TP53-deficient AML cells upregulate the mevalonate pathway upon CAR T-cell attack.
- CAR T-cells engaging TP53-deficient AML cells exhibit a downregulated Wnt pathway.
- Inhibition of the mevalonate pathway in TP53-deficient AML cells or enhancing the Wnt pathway in CAR T-cells restores efficient CAR T-cell-mediated AML cell lysis.

Please also suggest a striking image or visual abstract to illustrate your article as a PNG/jpeg/TIF file 550 px wide x 300-600 px high.

7/ As part of the EMBO Publications transparent editorial process initiative (see our Editorial at <http://embomolmed.embopress.org/content/2/9/329>), EMBO Molecular Medicine will publish online a Review Process File (RPF) to accompany accepted manuscripts.

This file will be published in conjunction with your paper and will include the anonymous referee reports, your point-by-point response and all pertinent correspondence relating to the manuscript. Let us know whether you agree with the publication of the RPF and as here, if you want to remove or not any figures from it prior to publication.

I look forward to receiving your revised manuscript.

Yours sincerely,

Lise Roth

**** Reviewer's comments ****

Referee #1 (Remarks for Author):

The authors took an extensive effort to address all the points I raised in the first round. The new results are of outstanding quality and clarity, and therefore further strengthen the manuscript.

I just have one minor remark to the TP53-deficient MV4;11 that has been newly generated in the course of this work. While the authors provide sufficient information about how this line was generated in the Supplementary Information, no results are shown to validate functional loss of the TP53 gene in these cells. The authors should provide some results that characterize the

mutations in the TP53 gene that were introduced and / or proof that the TP53 response is inactivated in these cells. Did authors use a single-cell-derived clone or a mutagenized cell pool for the experiments?

Referee #2 (Remarks for Author):

The authors have done an excellent job of addressing the prior concerns, and the revised manuscript is significantly improved. I would recommend publication.

**** Reviewer's comments ****

Referee #1 (Remarks for Author):

The authors took an extensive effort to address all the points I raised in the first round. The new results are of outstanding quality and clarity, and therefore further strengthen the manuscript.

I just have one minor remark to the TP53-deficient MV4;11 that has been newly generated in the course of this work. While the authors provide sufficient information about how this line was generated in the Supplementary Information, no results are shown to validate functional loss of the TP53 gene in these cells. The authors should provide some results that characterize the mutations in the TP53 gene that were introduced and / or proof that the TP53 response is inactivated in these cells. Did authors use a single-cell-derived clone or a mutagenized cell pool for the experiments?

Response: We thank the reviewer for this suggestion. We have indeed performed such validation experiments in the past and are happy to include them in the updated manuscript. For the experiments in this manuscript, we used a mutagenized polyclonal cell pool. After lentivirally-delivered CRISPR/Cas9-mediated gene editing and fluorescence-activated cell sorting, we performed functional validation of *TP53* deficiency via immunoblotting for p53 and p21 following treatment with typical p53-activating agents (Nutlin-3a and Daunorubicin; Appendix Figure S6A). Additionally, we measured apoptosis of CRISPRed cells upon treatment with Nutlin-3a for 72 hours (Appendix Figure S6B). In both experiments, we were able to demonstrate impaired p53 and p21 activation in these cell lines resulting in impaired apoptosis. Based on the clear functional evidence of p53 deficiency we therefore did not characterize mutations further.

Referee #2 (Remarks for Author):

The authors have done an excellent job of addressing the prior concerns, and the revised manuscript is significantly improved. I would recommend publication.

Response: We thank the reviewer for the time and effort spent on reviewing our manuscript and are happy that it regarded so favorably.

8th Jan 2024

Dear Prof. Boettcher,

Thank you for sending your revised files. I am pleased to inform you that your manuscript is accepted for publication and is now being sent to our publisher to be included in the next available issue of EMBO Molecular Medicine!

With kind regards,

Lise Roth
